# Possible poriferan body fossils in early Neoproterozoic microbial reefs

Elizabeth C. Turner[1✉]

Molecular phylogeny indicates that metazoans (animals) emerged early in the Neoproterozoic era[1], but physical evidence is lacking. The search for animal fossils from the Proterozoic eon is hampered by uncertainty about what physical characteristics to expect. Sponges are the most basic known animal type[2,3]; it is possible that body fossils of hitherto-undiscovered Proterozoic metazoans might resemble aspect(s) of Phanerozoic fossil sponges. Vermiform microstructure[4,5], a complex petrographic feature in Phanerozoic reefal and microbial carbonates, is now known to be the body fossil of nonspicular keratosan demosponges[6–10]. This Article presents petrographically identical vermiform microstructure from approximately 890-million-year-old reefs. The millimetric-to-centimetric vermiform-microstructured organism lived only on, in and immediately beside reefs built by calcifying cyanobacteria (photosynthesizers), and occupied microniches in which these calcimicrobes could not live. If vermiform microstructure is in fact the fossilized tissue of keratose sponges, the material described here would represent the oldest body-fossil evidence of animals known to date, and would provide the first physical evidence that animals emerged before the Neoproterozoic oxygenation event and survived through the glacial episodes of the Cryogenian period.

Benthic microbial structures (stromatolites and other microbialites) provide conspicuous evidence of pre-Phanerozoic life, but are difficult to understand because they rarely preserve recognizable evidence of the organisms involved. Stromatolitologists have struggled for over a century to decipher their microscopic laminae and clots, which are assumed to have been produced or influenced by in vivo and/or post-mortem biogeochemical activity, and to formalize the 'taxonomy' of their morphology and microstructure[5,11].

The existence of metazoans by the Ediacaran period (the last period of the Neoproterozoic) is indicated by bilaterian 'body' and trace fossils[12], and geochemical evidence (biomarkers)[13] provides disputed[14,15], indirect evidence for Cryogenian poriferans. The search for definitive physical evidence of pre-Cryogenian metazoans is confounded by uncertainty about what to look for, but preserved physical evidence should be small, subtle and possibly altogether unfamiliar. Given that sponges are the most basic of known animals[2,3], physical evidence of Neoproterozoic sponges could be sought, but effort focused on the characteristics of mineralized sponge skeletons (siliceous or calcareous spicules)[16–18] overlooks sponges with only proteinaceous (spongin or keratin[19,20]) skeletons. Early metazoan evidence might instead resemble taphonomic (preservational) products of sponge soft tissue[21–23] rather than mineralized sponge skeletal components. Although molecular clock data suggest that sponges emerged in the early Neoproterozoic[1], the oldest undisputed sponge body fossils are from the Cambrian period[15].

Recent work[6,7] has shown that vermiform microstructure[4,5]—an unusual microscopic feature in Phanerozoic reefs and stromatolites that was initially interpreted as being related to algae[4] or protozoans[24,25]—is instead a keratose sponge body fossil comprising complexly anastomosing cement-filled microtubules enclosed in carbonate microspar. It is produced taphonomically[6,10] in nonspicular keratose demosponges through post-mortem calcification of soft tissue to produce carbonate microspar (automicrite), which surrounds the tough spongin fibres of the 'skeleton' of the sponge. Decay of the spongin then produces a network of complexly anastomosing tubular moulds that eventually become passively filled with sparry calcite cement. Although questioned[26], the association between vermiform microstructure and sponges has been confirmed in undisputed body fossils of Phanerozoic sponges[7]. Three-dimensional reconstruction of vermiform microstructure has shown that tubule shape and branching configuration are too consistent and complex to be abiogenic (for example, compacted peloids), do not resemble the branching style of other possible organism types (microbial or fungal)[6] and are identical to the spongin meshworks of keratose sponges[6]. Although the existence of Proterozoic vermiform microstructure has been predicted[6,10,27], published examples are rare[28,29] and difficult to understand.

The calcification of decaying sponge soft tissue has been documented in modern sponges[21,22], and produces sponge 'mummies'[22] as well as a range of subtle carbonate sedimentary textures (such as peloid clusters) in living and Phanerozoic fossil sponges[7,9,21,22]. Taphonomic sediment textures (polymuds) that may be poriferan-related have previously been identified in the reefs that are the subject of this Article[23].

## Background

This petrographic study presents possible evidence of sponge body fossils in thin sections (30-μm-thick rock slices, viewed microscopically in

[1]Harquail School of Earth Sciences, Laurentian University, Sudbury, Ontario, Canada. ✉e-mail: eturner@laurentian.ca

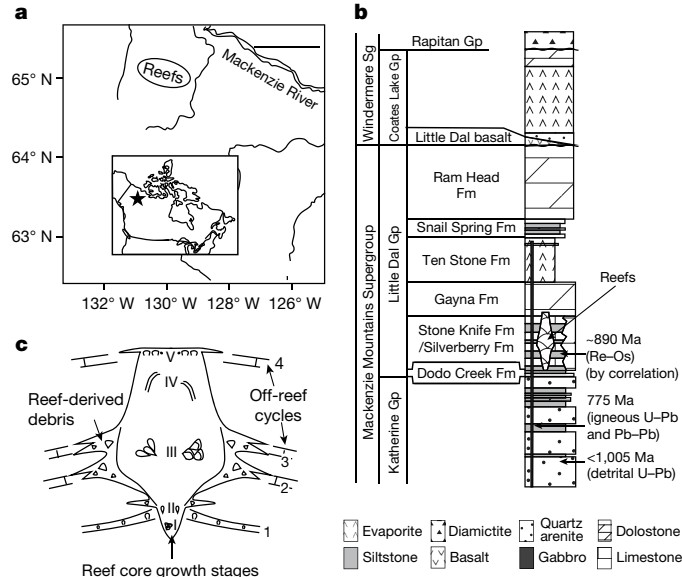

**Fig. 1 | Geographic and stratigraphic location of the study material. a**, Location of Little Dal reefs in northwestern Canada. Scale bar, 100 km. **b**, Stratigraphic position of the Little Dal reefs in the Stone Knife Formation (Fm)[32]; depositional age is known through litho- and chemostratigraphic correlation with the 892 ± 13-Ma (Re–Os black shale)[30] Boot Inlet Formation[31] (Shaler Supergroup (Sg)), together with other geochronological constraints[50,51]. Gp, group. **c**, Reef growth stages[34,35], simplified summary of framework morphologies[35] and off-reef cycles[34].

transmitted light) from the approximately 890-million-year-old (Ma)[30,31] Little Dal reefs (Stone Knife Formation[32], northwestern Canada) (Fig. 1a, b). These large (about 500 m in thickness, and kilometres in diameter) microbial reefs[33–36] were built mainly by variably preserved calcimicrobes that have been interpreted as filamentous cyanobacteria (photosynthesizers)[33,35,36], and developed palaeotopographic relief of up to about 100 m above the surrounding subphotic, level-bottom carbonate-mud seafloor. Reef framework, which is generally not discernible in natural exposures, was documented from slabbed hand samples and thin sections[35].

The reefs grew in five stages (Fig. 1c), each with different microbialite morphologies: anastomosing millimetre-to-centimetre-scale masses with no consistent shape (stages I–III); centimetre-scale anastomosing columns and digits (stages II, III and V); and steep sheet-like masses at a scale of decimetres to 10 m (stage IV). Stage V includes cement-rich to micritic, domical, turbinate and columnar stromatolites that generally lack calcimicrobes, with associated ooids and stromaclasts. Microbialites of stage I to IV grew predominantly in moderate-energy, illuminated palaeoenvironments[33,35,36], but stage V records a shallow-water, high-energy environment. The microstructure that forms most of the reefal microbialites (especially in stages III and IV) comprises filaments that are about 10 μm in diameter, separated by 10–100-μm masses of marine cement that probably represents the calcified sheath polysaccharide of cyanobacteria[33,36] that was permineralized during microbialite growth. The reef framework, consisting predominantly of this microstructure and its taphonomically degraded equivalents[36], defines primary void networks (millimetres to centimetres in size) that are commonly floored with geopetal carbonate mud and lined by isopachous, fibrous marine calcite cement. The relative timing of void-filling by marine cement precipitation versus geopetal sediment accumulation is variable, attesting to the very early timing of marine cement precipitation.

## Results

Vermiform microstructure in samples from stages II, III and V of the Little Dal reefs (Fig. 1c) is identifiable only in rare thin sections, in which

it forms millimetre-to-centimetre-scale masses of anastomosing tubes that are filled with calcite spar and surrounded by calcite microspar groundmass (Fig. 2a, b). The approximately 20–30-μm-wide tubules have complex, divergent branching and rejoining at a spacing of about 30–100 μm, form very irregular three-dimensional polygonal meshworks, are defined by enclosing microspar, lack walls, and are filled with clear, equant calcite crystals up to 20 μm wide (Fig. 2b). The homogeneous microspar groundmass that encloses the tubules comprises cloudy, equant, interlocking calcite crystals of about 2–8 μm wide, and differs texturally and compositionally from other fine-grained reefal carbonate in its uniform crystal size, lack of sedimentary texture, and dearth of detrital silicate impurities. Vermiform microstructure preservation is good to barely discernible.

Vermiform microstructure is present in three microfacies (i, ii and iii (the last divided into iiia and iiib subsets)), representing three palaeoenvironments (Figs. 2, 3). It is not present in calcimicrobe-dominated stromatolites or level-bottom carbonate mudstone that is distal to reefs.

In microfacies i (Fig. 2d, e, g, j, Extended Data Figs. 1, 2), vermiform microstructure is intercalated with carbonate mud (with or without larger reef-derived clasts and terrigenous impurities) in (1) synsedimentary debris flanking reef stages II and III, and (2) millimetre-to-metre-scale palaeodepressions on reef growth surfaces (stages II and III). It locally encrusts sides of reef-framework clasts in sediment, extends into crevices, and occupies shelter porosity under clasts (Fig. 2e).

In microfacies ii (Fig. 2f, Extended Data Fig. 3), vermiform microstructure occupies millimetre-to-centimetre-scale framework voids of reef stages II and III. Void-filling vermiform microstructure of microfacies i and ii either underlies (Fig. 2f) or overlies (Fig. 2e) isopachous void-lining marine cement.

In rare microfacies iiia (Fig. 2g–k, Extended Data Fig. 4), vermiform microstructure encrusts non-calcimicrobial microbialite columns and mingles with irregular muddy microbialite microstructure of reef stages II and III. In rare microfacies iiib (Extended Data Fig. 5), vermiform microstructure is sub-millimetrically interlayered with stage-V non-calcimicrobial stromatolites where it locally passes laterally to geopetal peloid accumulations in lenticular voids.

## Discussion

The shape, size, branching style and polygonal meshworks of the Little Dal vermiform tubules closely resemble both spongin fibre networks of modern keratosan sponges (Fig. 2a–c) and vermiform microstructure either demonstrated or interpreted to be sponge-derived in diverse Phanerozoic microbial, reefal and non-reefal carbonate rocks[6–8,10,24,25,27,37]. The compositional and textural homogeneity of the microspar groundmass supports an origin through permineralization of a pre-existing biological substance[9], rather than incremental accumulation of detrital sediment or microbial carbonate that passively incorporated complexly anastomosing tubular microfossils. Variable preservation and association with geopetal peloid accumulations are familiar aspects of Phanerozoic sponge taphonomy[9,21,22,38]. In previous work, detailed comparison of the three-dimensional characteristics of vermiform microstructure with branching cylindrical organism types yielded no convincing alternative to the sponge interpretation[6].

The preference of Little Dal vermiform microstructure for environments that were not inhabited by photosynthetic calcimicrobes (reef flanks, depressions on active reef growth surface, and framework and shelter voids), versus its absence from filamentous calcimicrobial reef-framework components, suggests that (1) illumination may not have been a requirement and (2) the organism may have been unable to compete with reef-building photosynthesizers that grew and/or calcified rapidly. The interlayering of vermiform microstructure with calcimicrobe-free microbialite (microfacies iiib) in the high-energy, well-illuminated reef surfaces of reef stage V supports the

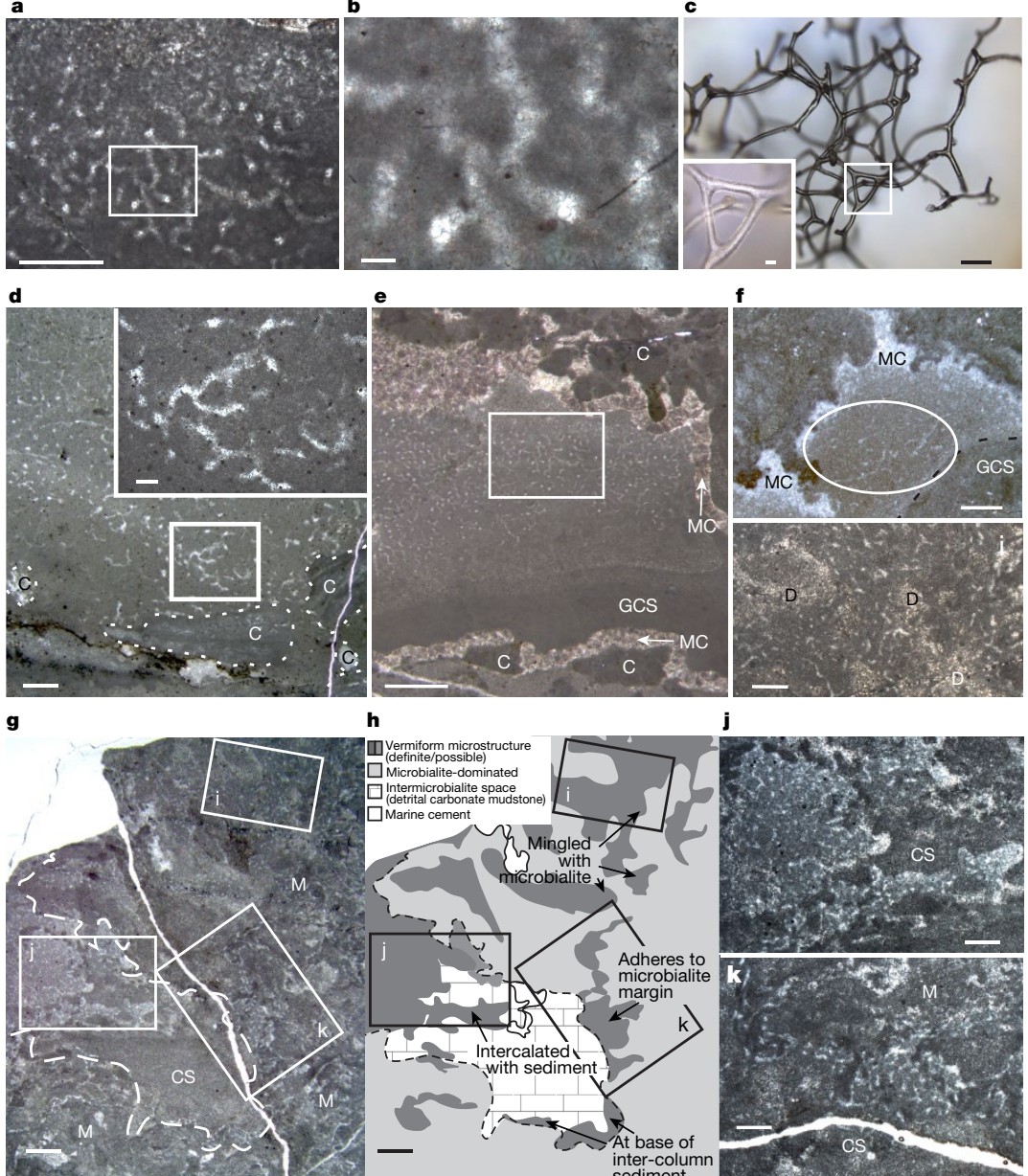

**Fig. 2 | Characteristics and distribution of Little Dal vermiform microstructure in stratigraphically oriented 30-μm-thick thin sections.**
**a**, Well-preserved vermiform microstructure exhibits a polygonal meshwork of anastomosing, slightly curved, approximately 30-μm-diameter tubules embedded in calcite microspar (KEC25). Scale bar, 500 μm. **b**, Enlarged rectangle from **a**, showing branching tubules forming three-dimensional polygons intersected at various angles by the thin section; clear calcite crystals, about 10–20 μm in width, fill tubules in groundmass of more finely crystalline calcite (dark grey). Scale bar, 50 μm. **c**, Three-dimensional fragment of spongin skeleton from a modern keratosan sponge, illustrating its branching and anastomosing network of fibres (incident light). Scale bars, 100 μm (main panel), 20 μm (inset). **d**, Vermiform microstructure in debris that includes calcimicrobialite and other reef-derived clasts (C) flanking reef stage III (MV63). Scale bars, 1 mm (main panel), 100 μm (inset). **e**, Vermiform microstructure in shelter pore beneath microbialite clast, in detrital sediment occupying a reef-top depression; pore is thinly lined with marine calcite cement (MC) (indicated with an arrow), and partly filled with geopetal carbonate sediment (GCS) (KEC25; stage-III reef core). Rectangle is enlarged in **a**. Scale bar, 1 mm. **f**, Vermiform microstructure in a microbialite (M) framework

void is overlain by pore-occluding marine calcite cement; circled area indicates moderately well-preserved tubule meshwork (DL32a; reef stage II; detailed characteristics depicted in Extended Data Fig. 1). Scale bar, 1 mm. **g**, Patches of vermiform microstructure in various relationships with micritic microbialite masses (white dashed outline) and detrital carbonate sediment (CS) (KES23; resedimented stage-II reef clast). Rectangles are enlarged in **i**–**k**. Scale bar, 1 mm. **h**, Simplified depiction of relationships among vermiform microstructure, microbialite masses and detrital carbonate sediment in **g**. Scale bar, 1 mm. **i**, Vermiform microstructure mingled with microbial micrite within a microbialite digit (enlarged from **g**). Scale bar, 500 μm. **j**, Vermiform-microstructured mass within sediment between microbialite digits; also contains diagenetic dolomite patches (D) (enlarged from **g**). Scale bar, 500 μm. **k**, Vermiform-microstructured mass adhering to the margin of microbialite digit (enlarged from **g**). Scale bar, 500 μm. All images except **c** are in plane-polarized transmitted light. Samples from resedimented reef debris are depicted in depositional orientation based on geopetal structures. Reef locations and abbreviations (such as KEC) are described in a previous publication[35]. Larger versions of vermiform microstructure photomicrographs are provided in Extended Data Figs. 1–5.

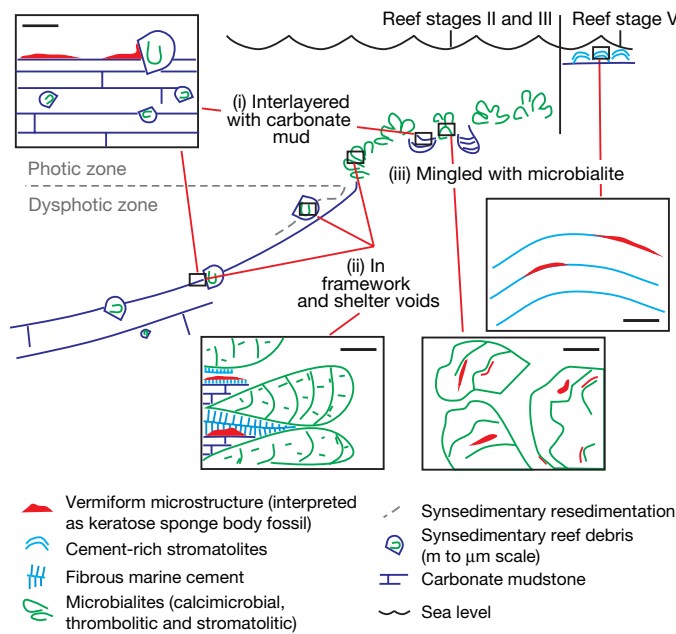

**Fig. 3 | Palaeoenvironments occupied by the Little Dal vermiform microstructure interpreted as possible body fossils of keratose sponges.** The organism lived (i) on poorly illuminated to non-illuminated carbonate mud surfaces in depressions on the reef surface and on debris aprons mantling reef flanks; (ii) in voids produced by the growth of the complex microbial framework of the reef; and (iii) interlayered with non-calcimicrobial microbialites (cement-rich and muddy-laminated stromatolites in high-energy reef-capping phase V; irregularly muddy-laminated to clotted microbialites in moderate-energy environments of reef stages II and III). Scale bars, 5 mm.

interpretation that the vermiform-microstructured organism was not capable of competing with reef-building filamentous cyanobacteria, but instead occupied niches in which the filamentous calcimicrobes did not live owing to (1) poor illumination or (2) high hydrodynamic energy. The occupation of cryptic microniches (shelter and reef framework voids) by sponges (for example, microfacies i and ii), is well known in the Phanerozoic[21,37,39,40].

The obligatory spatial association of vermiform microstructure with reefs built by oxygen-producing cyanobacteria may indirectly support a metazoan interpretation. Prior to the Neoproterozoic oxygenation event, marine dissolved oxygen was probably low[41] except perhaps in the vicinity of photosynthesizing microbial communities; the metabolic requirements of metazoans may have limited early animals to localized, comparatively well-oxygenated (for the time) environments (oxygen 'oases'). Given the approximately 890-Ma depositional age[30,31], the vermiform-microstructured Little Dal organism may been tolerant of 'low' oxygen (that is, relative to modern levels), which is a characteristic of some modern and fossil sponges[42].

If the vermiform-microstructured masses in the Little Dal reefs are accepted as early sponge body fossils, their approximately 890-Ma age would imply that (1) the evolutionary emergence of metazoans was decoupled from the Neoproterozoic oxygenation event[41–45] and (2) early animal life was not catastrophically affected by the Neoproterozoic glacial episodes. If the Little Dal objects are truly sponge body fossils, they are older than the next-youngest undisputed sponge body fossils (Cambrian)[15] by approximately 350 million years.

It would not be surprising to find that the earliest sponges were reef-dwellers; the history of Phanerozoic reefs is rich with reef-building and reef-dwelling sponges[46]. If the masses of vermiform microstructure in the Little Dal reefs were to be accepted as an early Neoproterozoic expression of sponge tissue preservation, their age and proposed identity would be compatible with (1) evidence that the opisthokont

(animal + fungus) clade was already established by the time of the Meso-proterozoic–Neoproterozoic transition[47,48], (2) possible evidence of 1-billion-year-old multicellular holozoans[49], (3) molecular clock estimates for the emergence of the Porifera in the early Neoproterozoic[1] and (4) a revised taxonomy of nonspiculate keratose sponges showing that they are a sister group to other demosponges[19]. The Little Dal vermiform microstructure is perhaps exactly what should be expected of the earliest metazoan body fossils: preservation through post-mortem calcification of sponge-grade soft tissue in the decaying bodies of small, shapeless, sessile, epibenthic and cryptic animals most closely affiliated with keratose sponges.

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

## Methods

Field work was done on foot from two-person, backpacking-style camps placed at sites that are accessible only by helicopter. Samples were collected at various times between 1992 and 2018, under all required permits. Recording sample locations using GPS is not possible for most sites owing to the extreme topography of the exposures' cliffs, pinnacles and canyons, and so sample location was documented using photographs and sketches. Several samples are from a mineral-exploration drill-core stored on-site in the field. Owing to the homogeneous grey weathering of reef surfaces, lithofacies cannot be identified in the field. Instead, hand samples were collected and later slabbed and thin-sectioned. Vermiform microstructure was identified in a small proportion of the samples collected. Repeat visits focused primarily on resampling the rare areas in which vermiform microstructure had been identified.

Standard 30-µm-thick thin sections were examined in plane-polarized transmitted light using a Nikon C-Pol binocular microscope fitted with digital camera and Luminera Infinity Analyze software (for lower-magnification images) and an Olympus BX-51 petrographic microscope equipped with Q-Imaging digital capture system (for higher-magnification images).

**Reporting summary**

Further information on research design is available in the Nature Research Reporting Summary linked to this paper.

## Data availability

All relevant data are contained with the Article and its Supplementary Information, or are available from the author upon reasonable request.

**Acknowledgements** Field work and sample preparation were supported by Natural Sciences and Engineering Research Council (NSERC) Discovery Grants (E.C.T.), and the Agouron Institute (2016), based on locations initially discovered during post-graduate work on microbialites funded through the NSERC grants of G. M. Narbonne and N.P. James.

**Author contributions** E.C.T. conducted all aspects of the study.

**Competing interests** The author declares no competing interests.

**Additional information**
**Correspondence and requests for materials** should be addressed to E.C.T.

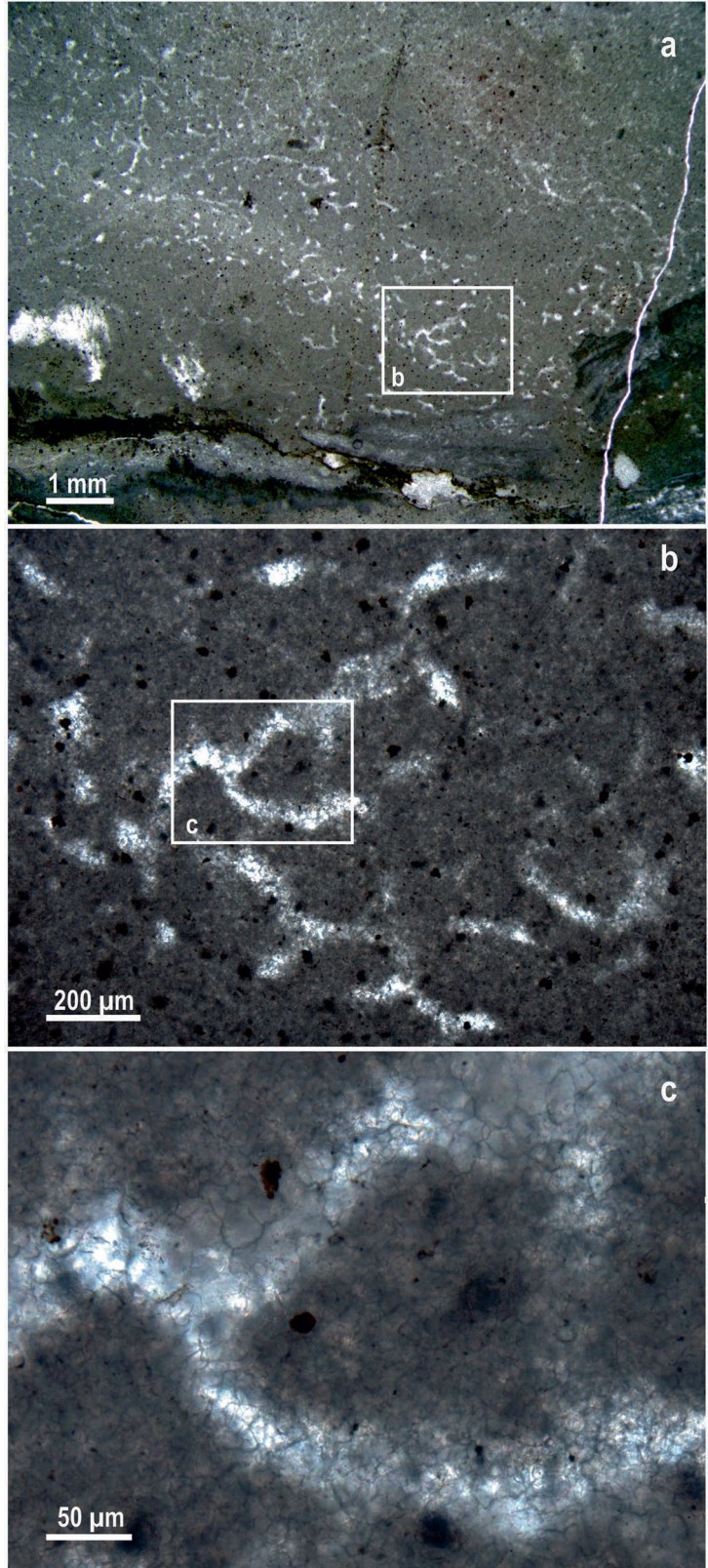

**Extended Data Fig. 1 | Vermiform microstructure of microfacies i.**
Elaboration of Fig. 2d. **a**, Vermiform microstructure is intercalated with detrital sediment (lower 1/3 of image) that includes calcimicrobialite clasts.
**b**, Three-dimensional meshwork of vermiform microstructure transected by the plane of the 30-μm-thick thin section shows anastomosing tubule system; enlarged from **a**. **c**, Anastomosing tubules occluded by clear blocky calcite enclosed by cloudy calcite groundmass of slightly smaller crystals; enlarged from **b**. The tubules have no constructional walls and are defined by the dark enclosing groundmass. Oriented sample MV63 in plane-polarized transmitted light.

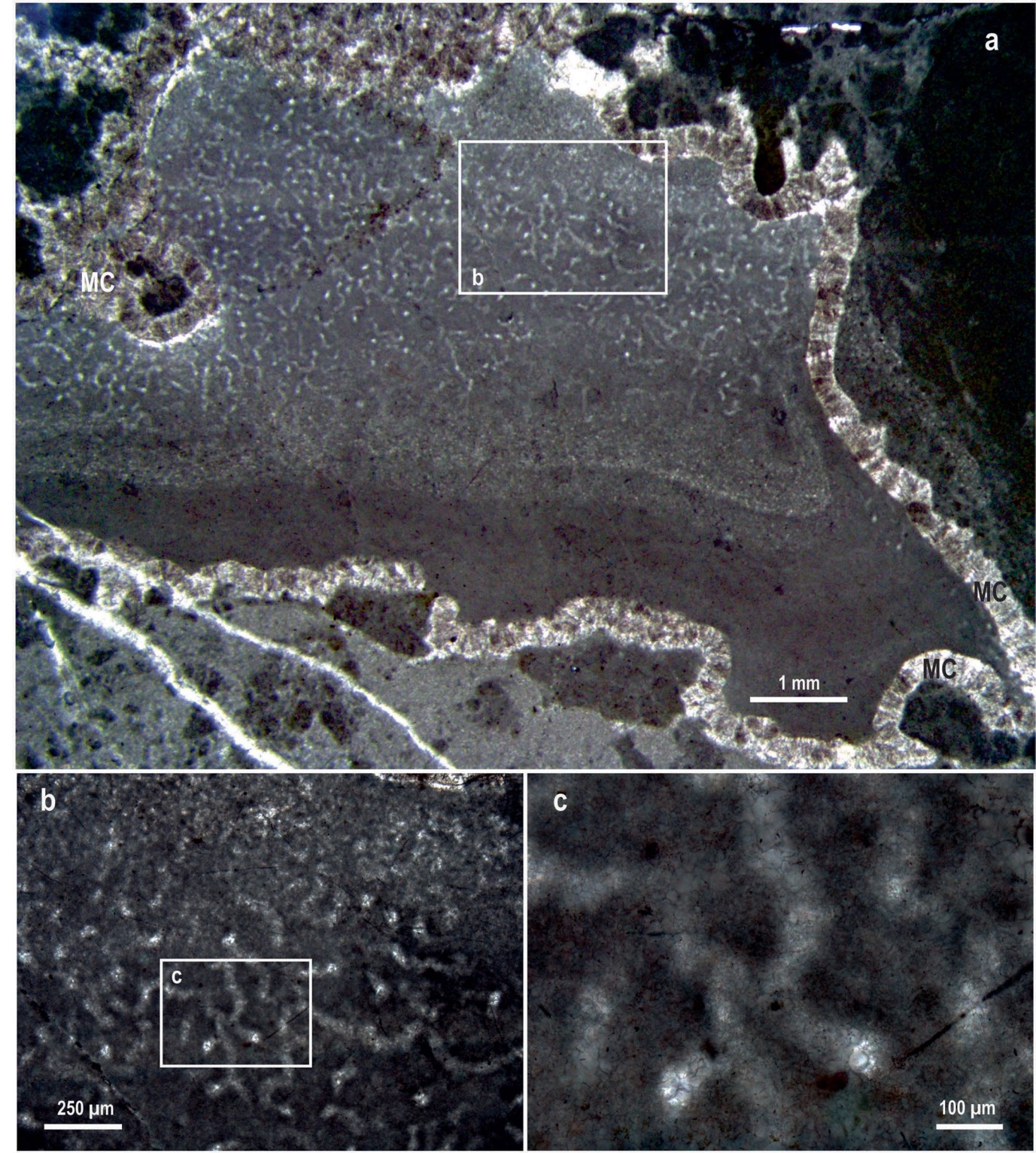

**Extended Data Fig. 2 | Vermiform microstructure of microfacies i.**
Elaboration of Fig. 2e. **a**, Vermiform microstructure in a shelter void beneath an irregular microbialite clast and floored by angular reef clasts in carbonate mud, from a reef-top depression. Fibrous isopachous marine calcite cement lined the pore before its occupation by the vermiform mass. **b**, Three-dimensional meshwork of vermiform microstructure transected by the plane of the 30-µm-thick thin section shows anastomosing tubule system; enlarged from **a**. **c**, Anastomosing tubules occluded by clear blocky calcite enclosed by cloudy calcite groundmass of slightly smaller crystals; enlarged from **b**. The tubules have no constructional walls and are defined by the dark enclosing groundmass. Oriented sample KEC25 in plane-polarized transmitted light.

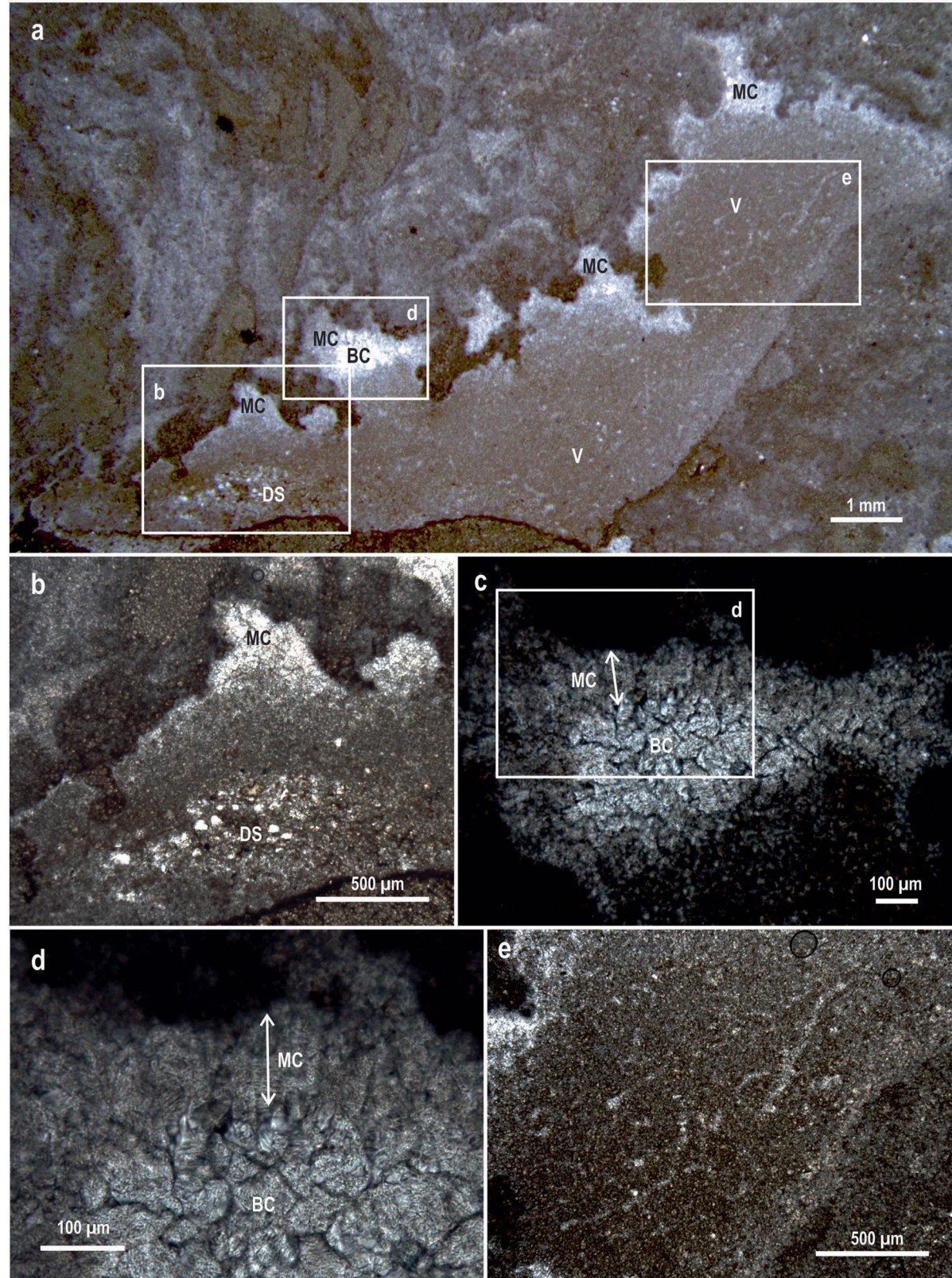

**Extended Data Fig. 3 |** See next page for caption.

**Extended Data Fig. 3 | Spatial–temporal relationships among vermiform microstructure, geopetal detrital sediment and marine cement that collectively fill reef framework voids of microfacies ii.** Elaboration of Fig. 2f. **a**, Reef framework void among non-calcimicrobial stromatolites of reef stage II. Detrital sediment (DS) occupies lowest part of the void. Much of the void is occupied by vermiform microstructure (V); uppermost parts of the void are occupied by marine cement (MC) and local burial cement (BC). **b**, Detrital sediment accumulation, which includes quartz silt (transparent white particles); enlarged from **a**. **c**, Upper part of void is lined by isopachous cloudy marine calcite cement (arrow) (MC) and the remaining porosity occluded by burial calcite cement (pale) (BC); enlarged from **a**. **d**, Enlargement from **c** to demonstrate cloudy, isopachous, fibrous nature of the marine calcite cement, versus more transparent, equant, blocky shape of burial cement that occupies the small amount of pore space remaining after accumulation of geopetal sediment, growth of vermiform microstructure and precipitation of marine cement. **e**, Moderately preserved vermiform microstructure; enlarged from **a**. Oriented sample DL32a from reef stage II, in plane-polarized transmitted light.

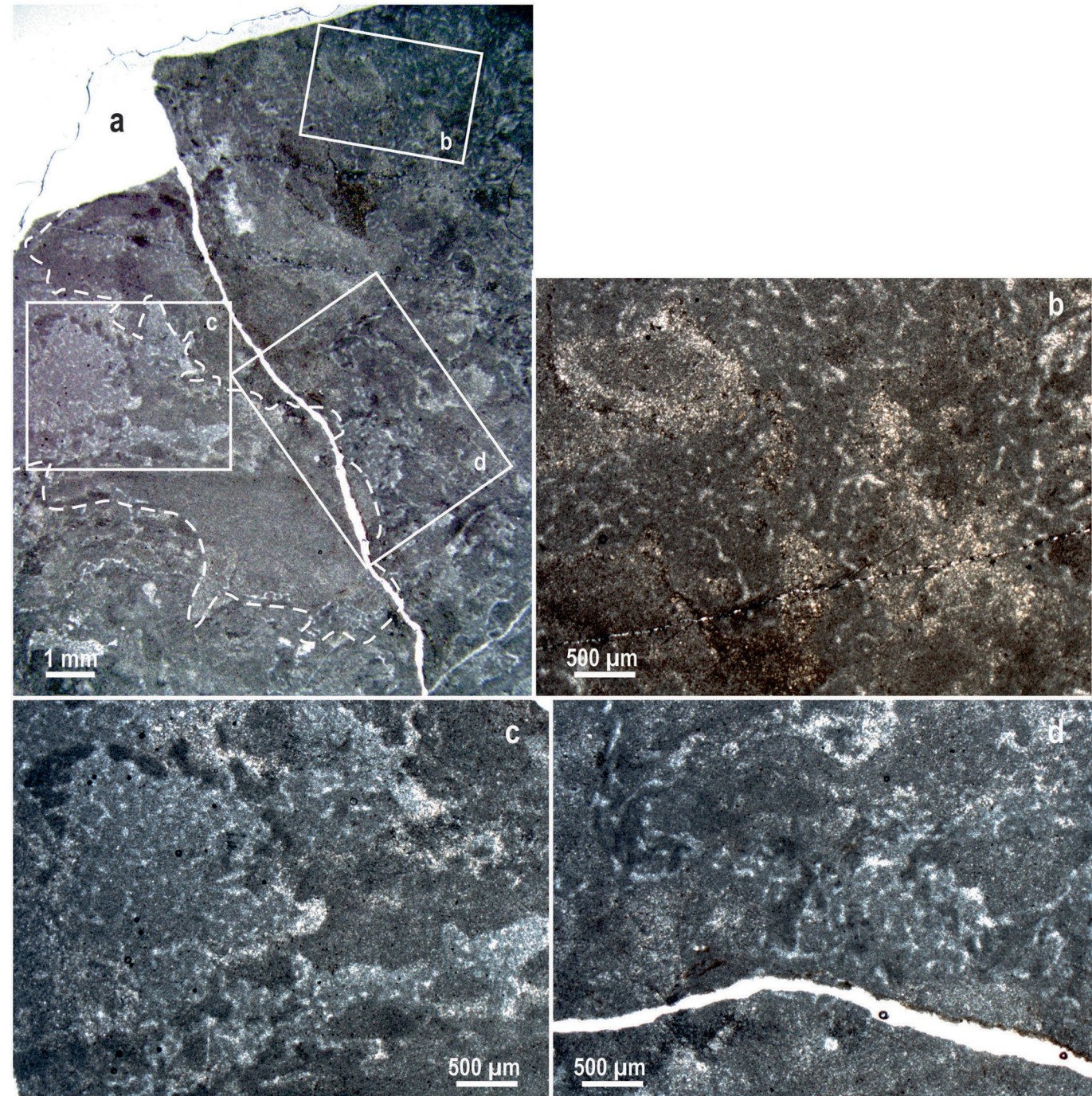

**Extended Data Fig. 4 | Associations of vermiform microstructure, micritic microbialite columns and detrital sediment in microfacies i and iiia.** Elaboration of Fig. 2g–k. **a**, Low-magnification image of two irregular microbialite columns and inter-column sediment (outlined by white dashed line). **b**, Vermiform microstructure is intermingled with muddy microbialite in column interior; enlarged from **a**. **c**, Vermiform microstructure intercalated with inter-column sediment; enlarged from **a**. **d**, Vermiform microstructure encrusting the margin of a microbialite column; enlarged from **a**. Sample KES23, from reef stage II, in plane-polarized transmitted light; sample is from a resedimented reef clast and is oriented on the basis of geopetal structures out of the field of view.

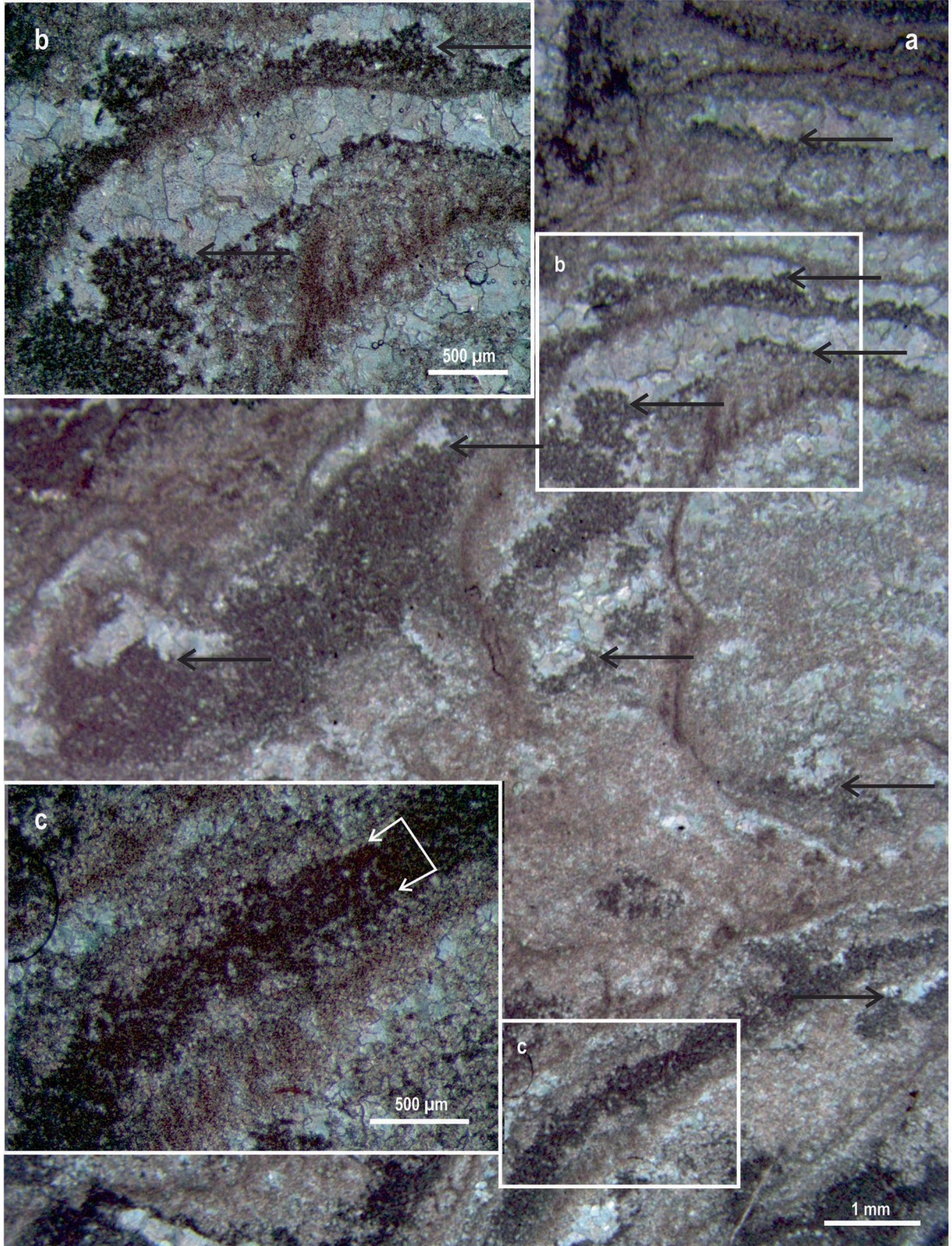

**Extended Data Fig. 5 | Spatial and textural association of vermiform microstructure and geopetal peloids in microfacies iiib.** This association suggests that geopetal peloid accumulations are a taphonomic product derived through poor preservation of vermiform microstructure. **a**, Poorly preserved vermiform microstructure (arrowed; inset c) adjacent to laminar to lenticular voids containing geopetal peloid accumulations (arrowed; inset b) in cement-rich non-calcimicrobial stromatolites of reef stage V. Oriented IR drillcore sample RT A19-495.4′ in plane-polarized transmitted light.

# nature research

| | |
|---|---|

# Reporting Summary

Nature Research wishes to improve the reproducibility of the work that we publish. This form provides structure for consistency and transparency in reporting. For further information on Nature Research policies, see Authors & Referees and the Editorial Policy Checklist.

## Statistics

For all statistical analyses, confirm that the following items are present in the figure legend, table legend, main text, or Methods section.

| n/a | Confirmed | |
|---|---|---|
| ☒ | ☐ | The exact sample size (*n*) for each experimental group/condition, given as a discrete number and unit of measurement |
| ☒ | ☐ | A statement on whether measurements were taken from distinct samples or whether the same sample was measured repeatedly |
| ☒ | ☐ | The statistical test(s) used AND whether they are one- or two-sided *Only common tests should be described solely by name; describe more complex techniques in the Methods section.* |
| ☒ | ☐ | A description of all covariates tested |
| ☒ | ☐ | A description of any assumptions or corrections, such as tests of normality and adjustment for multiple comparisons |
| ☒ | ☐ | A full description of the statistical parameters including central tendency (e.g. means) or other basic estimates (e.g. regression coefficient) AND variation (e.g. standard deviation) or associated estimates of uncertainty (e.g. confidence intervals) |
| ☒ | ☐ | For null hypothesis testing, the test statistic (e.g. *F*, *t*, *r*) with confidence intervals, effect sizes, degrees of freedom and *P* value noted *Give P values as exact values whenever suitable.* |
| ☒ | ☐ | For Bayesian analysis, information on the choice of priors and Markov chain Monte Carlo settings |
| ☒ | ☐ | For hierarchical and complex designs, identification of the appropriate level for tests and full reporting of outcomes |
| ☒ | ☐ | Estimates of effect sizes (e.g. Cohen's *d*, Pearson's *r*), indicating how they were calculated |

*Our web collection on statistics for biologists contains articles on many of the points above.*

## Software and code

Policy information about availability of computer code

| | |
|---|---|
| Data collection | Photomicrographs were imaged using Luminera Infinity Analyze v.6.5 and Qcapture 2.98.0. |
| Data analysis | No software was used for data analysis. |

For manuscripts utilizing custom algorithms or software that are central to the research but not yet described in published literature, software must be made available to editors/reviewers. We strongly encourage code deposition in a community repository (e.g. GitHub). See the Nature Research guidelines for submitting code & software for further information.

## Data

Policy information about availability of data

All manuscripts must include a data availability statement. This statement should provide the following information, where applicable:
- Accession codes, unique identifiers, or web links for publicly available datasets
- A list of figures that have associated raw data
- A description of any restrictions on data availability

No databases were generated in or used by this study.

# Field-specific reporting

Please select the one below that is the best fit for your research. If you are not sure, read the appropriate sections before making your selection.

☐ Life sciences    ☐ Behavioural & social sciences    ☒ Ecological, evolutionary & environmental sciences

For a reference copy of the document with all sections, see nature.com/documents/nr-reporting-summary-flat.pdf

# Ecological, evolutionary & environmental sciences study design

All studies must disclose on these points even when the disclosure is negative.

| | |
|---|---|
| Study description | Describes petrographic evidence of possible body fossils of sponges that are approximately 890 million years old. |
| Research sample | Thin sections (30-micon-thick rock slices) from fossil reef rock in the Stone Knife Formation ("Little Dal reefs") in Northwest Territories, Canada. |
| Sampling strategy | Over a thousand approximately fist-sized rock samples were separated from natural rock exposures using a rock hammer. The samples were later slabbed and thin-sectioned. Initially, samples had been collected for an unrelated purpose (documenting reef microbialites in a separate, published study); sample distribution was randomly dispersed throughout all exposed parts of selected reefs in an attempt capture spatial variability in microbialites, which are not discernible on natural rock exposures. Sample field locations were documented using photographs and diagrams; obtaining accurate GPS points for sample locations is not possible given the extreme topography (limits satellite access) and the small size of the samples relative to GPS error. Areas where the thin sections contained vermiform microstructure were later revisited and resampled. Sample size is considerably larger than the masses of vermiform microstructure. Sample distribution in the reefs is dense enough and reef facies well enough understood (previously published study) for the reefal subenvironments in which vermiform microstructure is preserved to be characterised. |
| Data collection | Rock samples were separated from natural rock exposures using a rock hammer. Samples were shipped to the lab, sawed, polished, and thin-sectioned using standard petrographic preparation. |
| Timing and spatial scale | Rock samples were collected during field work between 1992 and 2018. Sample size is considerably larger than the masses of vermiform microstructure that are the subject of the study. |
| Data exclusions | No data were excluded. |
| Reproducibility | Reproducibility depends on locating the exact field locations and places on exposure surfaces from which samples containing vermiform microstructure were collected. Field locations of rock samples were recorded in detail using photographs and diagrams. Revisiting and resampling these locations in the years following the initial collection successfully yielded more material containing vermiform microstructure in thin section. |
| Randomization | This was not an experimental study. |
| Blinding | This was not an experimental study. |

Did the study involve field work?   ☒ Yes   ☐ No

## Field work, collection and transport

| | |
|---|---|
| Field conditions | Remote alpine-subarctic locations in the Mackenzie Mountains, Northwest Territories, Canada, that are accessible only by helicopter. Field work is possible only in the summer months when snow cover is minimal (mid-June to mid-August). |
| Location | Numerous locations between 64°47'N / 129°35'W and 64°59'N / 130°55'W. |
| Access and import/export | Field work was conducted under science licencing from the Aurora Research Institute (2016 Licence #15888; 2017, 2018 Licence #15993), and associated permissions from land use, water board, renewable resource, community, land claim, band council, and hunting-trapping organisations. |
| Disturbance | No temporary nor long-term disturbances were introduced. |

# Reporting for specific materials, systems and methods

We require information from authors about some types of materials, experimental systems and methods used in many studies. Here, indicate whether each material, system or method listed is relevant to your study. If you are not sure if a list item applies to your research, read the appropriate section before selecting a response.

### Materials & experimental systems

| n/a | Involved in the study |
|---|---|
| ☒ | ☐ Antibodies |
| ☒ | ☐ Eukaryotic cell lines |
| ☐ | ☒ Palaeontology |
| ☒ | ☐ Animals and other organisms |
| ☒ | ☐ Human research participants |
| ☒ | ☐ Clinical data |

### Methods

| n/a | Involved in the study |
|---|---|
| ☒ | ☐ ChIP-seq |
| ☒ | ☐ Flow cytometry |
| ☒ | ☐ MRI-based neuroimaging |

## Palaeontology

| | |
|---|---|
| Specimen provenance | Field work was conducted under science licencing from the Aurora Research Institute (2016 Licence #15888; 2017, 2018 Licence #15993), and applicable associated permissions from land use, water board, renewable resource, community, land claim, band council, and hunting-trapping organisations. |
| Specimen deposition | Field data, rock samples, and thin sections are archived in the author's collection at Laurentian University. |
| Dating methods | No new dates are presented. |

☐ Tick this box to confirm that the raw and calibrated dates are available in the paper or in Supplementary Information.

