## [Peer Review File · Nature]

Manuscript Title: Possible poriferan body fossils in ca. 890 Ma (Tonian) microbial reefs

Reviewer Comments & Author Rebuttals

Reviewer Reports on the Initial Version:

Referees' comments:

Referee #1 (Remarks to the Author):

A. Summary of the key results

- Report of the earliest possible sponge remains, which may extend the fossil occurrence of sponges for 350 Myr or longer.

B. Originality and significance: if not novel, please include reference

- It is original and significant.

C. Data & methodology: validity of approach, quality of data, quality of presentation

- Data is generally OK, but can be improved. I suggest including some more data in a supplementary file. Details are provided below.

D. Appropriate use of statistics and treatment of uncertainties

- Not applicable

E. Conclusions: robustness, validity, reliability

- Valid conclusions.

F. Suggested improvements: experiments, data for possible revision

- Some more data can be provided. Details are provided below.

G. References: appropriate credit to previous work?

- Mostly OK, some can be improved. Details are provided below.

H. Clarity and context: lucidity of abstract/summary, appropriateness of abstract, introduction and conclusions

- All appropriate.

This paper shows possible sponge body fossils – the vermiform fabric – from the Tonian reefs of Canada and proves what I had predicted in my recent articles (Lee and Riding, 2021a, b). I love this paper, as this result will possibly extend the fossil occurrence of sponges for 350 Myr or longer. As the earliest sponges are among the most important topics among researchers of various disciplines, including paleontology, molecular phylogeny, geochemistry, etc., I think this paper will certainly be of immediate interest to many scientists and provoke discussion among them. I thus believe this manuscript is suitable to be published in Nature. I only have some minor comments that may improve the manuscript.

Line 34: Bilaterian 'body' and trace fossils, as indicated by Chen et al. (2019).

Line 62-63: Polymud fabric is reported from the same rock that the author studied (Neuweiler et al., 2009). I suggest moving this sentence to the discussion section to support the sponge origin of Little Dal vermiform fabric.

Lines 65-75 and onwards: Readers will be interested to know some more details of the reef. I know that the reef was extensively described by the author (Turner et al., 1997, 2000a, b), but it will be good to provide some summary here, as this is a separate manuscript. What are stages 1 and 4, for example? What kind of morphologies can be found there? More illustrations describing the reef growth stage will also be helpful. Figure 3 is a nice model, but it is not supported by the field/slab photo. These can be provided separately in the supplementary material.

Lines 97-98: It seems to me that there are some remains of calcimicrobes, e.g., Epiphyton or Angusticellularia, in Fig. 2F, although I cannot confidently identify the structure due to the low resolution of the photomicrograph.

Lines 99-104: It seems that preservation of microfacies 1 is worse than that of microfacies 2 or 3, as vermiform fabric fades away from the center to the margin of Fig. 2C. Also, branching tubules occur along the clast. These characteristics are quite similar to Park et al. (2015, *Sedimentary Geology*, v. 319, p. 124-133) or Lee et al. (2016, *Palaeogeography Palaeoclimatology Palaeoecology*, v. 457, p. 23-30).

Lines 110-113: Geopetal peloidal accumulations are well figured in Lee et al. (2016).

Lines 110-115: Is there any necessity to separate these two paragraphs?

Line 120: I don't think citing Luo et al. (2016) or Park et al. (2017) are appropriate here. These two references interpret vermiform microstructures as sponges, but except for that, they do not provide any firm evidence that vermiform microstructures have resulted from sponges. Luo et al. (2014) provided 3D reconstructions of the vermiform fabric and compared these with modern keratose sponges, and it is appropriate to be cited here. On the other hand, Lee et al. (2014, *Palaios*, v. 29, p. 27-37, fig. 7a-c) showed that vermiform microstructures can form definitive sponge outlines and is a better reference to be cited here.

Line 121: I don't think using apostrophe is necessary when quoting the word keratolite.

Line 137: Poor illumination can be a cause of the absence of cyanobacteria, but I don't think cyanobacteria can be inhibited by high hydrodynamic energy. Phanerozoic calcimicrobes are commonly found in high-energy shoal environments. See Lee and Riding (2021, *Geobiology*) for discussion.

Lines 154-163: One thing to be noted here is that non-spiculate demosponges, including Keratosa and Verongimorpha (sensu Erpenbeck et al., 2012, *Molecular Phylogenetics and Evolution*, v. 63, p. 809-816), are sister groups to all other spiculate demosponges (Worheide et al., 2012, *Advances in Marine Biology* book, p. 1-78), so maybe primitive sponges could have devoid of spicules. It will be interesting to discuss this idea here, which may be helpful for future studies.

Interestingly, Homoscleromorpha – the fourth class of sponges – is closer to the Calcarea than Demospongiae, although Homoscleromorpha has siliceous spicules and spongin fibers similar to demosponges which Calcarea lack. This may indicate the paraphyletic nature of sponges (Calcarea–Homoscleromorpha vs. Demospongiae–Hexactinellida) (Nosenko et al., 2013, *Molecular Phylogenetics and Evolution*, v. 67, p. 223-233), and, possibly that spicules may have appeared more than once during the sponge evolution. If the author has any problem with this comment, please feel free to contact me.

Fig. 1C. Please mention that <1005 Ma age is from detrital zircon (Leslie, 2009). If I understood correctly, Milton et al. (2017) analyzed Little Dal Basalt that overlies Little Dal Group (their fig. 2) using U-Pb, not Pb-Pb, and its age is 775 Ma, not 778 Ma. According to Milton et al. (2017, fig. 2), the gabbro dike that the author pointed is connected to the Little Dal Basalt. Please modify this figure accordingly.

Fig. 2E. I cannot clearly identify geopetal peloidal fabrics here, although I can presume that the fabric does exist. It will be good to provide a better photomicrograph.

Fig. 2F. I can identify vermiform fabrics here, but with provided photomicrograph, it will be hard to recognize vermiform fabrics for non-specialists due to the low resolution of the photomicrograph. I suggest providing a photomicrograph with a higher resolution.

Fig. 3. It will be good if (a), (b), (c) can be marked on the figure. phase5 -> phase 5. I found similar problems a couple of times (e.g., in Fig. 2 caption, line 68, line 119).

Referee #2 (Remarks to the Author):

The search for a fossil record of metazoans in rocks older than the Ediacaran has had a long, and fraught, history. There have been claims (e.g. Maloof et al. 2010), but none have been upheld to date. Many fossils are now known from the terminal Ediacaran (ca. 560 – 540 million years ago (Ma)) that are almost certainly metazoans, but the precise affinities of nearly all remains problematic. Molecular phylogenies predict that metazoans probably had an origin in the Tonian, and as sponges are sister-group to all other multicellular animals, there has been an expectation that such fossils could be found.

This submission proposes that microscopic vermiform structures, found associated with Tonian microbial reefs ca. 890 Ma, are in fact the remains of keratosan sponges – those with an organic, rather than calcified, skeleton. This is an extremely bold claim. This interpretation is based, in part, on the finding of similar structures intergrown with microbialite (including stromatolitic) reef frameworks that are fairly widespread in the Phanerozoic (e.g. Luo and Reitner, 2016; Lee and Riding, 2021). All Phanerozoic examples are found as dominantly encrusting structures, intergrowing with and attached to, microbialite surfaces. Such vermiform structures can therefore be shown to have grown upwards away from these substrates.

The Tonian structures presented here are proposed to have three expressions: intergrown with microbialite (Fig. 2E,F), within microbialite reef cavities (Fig. 2D), and as reworked clasts (Fig. 2C). But the examples found in reef cavities - which are by far the best preserved - are associated with sediment infill - mainly recrystallized micrite. They are not in any way attached to the reef framework. Most importantly, this sediment infill postdates the precipitation of early marine cements around the reef cavities – see Fig. 2D. So whatever these structures are, they therefore post-dated the active growth of the reef – they are not attached body fossils. Similar claims have been made for vermiform structures within Phanerozoic micrite-filled cavities to be sponges (e.g. Park et al., 2017), but this assertion remain controversial.

Figs. 2E,F and 2C are simply of too low resolution and possibly also poor preservation to really demonstrate that these structures are either intergrown with microbialite, or reworked clasts, respectively. To persuade the reader, far more convincing evidence needs to be presented.

There is much speculation in the Discussion as to the role of oxygen, etc., but this is all rather superfluous. The main claim that these Tonian vermiform structures are indeed sponges needs to be far better illustrated and described, to be believable.

Referee #3 (Remarks to the Author):

Elizabeth Turner presents a very intriguing paper dealing with the very early record (Tonian) of a sponge fossil. Based on molecular reconstructions the origin of animals/sponges should have happened in this time. However, a proof based on fossils was up to now not possible. Also the occurrence of chemical fossils (biomarker), e.g. 24-isopropylcholestane, from this time and a bit later are heavily discussed and problematic. Spicule record of sponges started in the base of the early Cambrian. The basic sponge type was non-spicular and probably related with the “Keratosa” an old taxonomic name of non-spicular sponges with an organic skeleton only. Modern phylogenetic reconstructions have shown that this group is split into the Myxospongia with an organic skeleton formed mainly of chitin and the “Keratosa” with an organic skeleton formed of spongin, a complex proteinaceous material. In any case both types of organic skeletons are very resistant against microbial degradation. In the end the molds of the skeleton fibers are cemented mainly by carbonate and form “vermiculate” fabrics. These sponges dwell in a wide range of ecological niches and often in dark cryptic areas.

The paper is well written and represents the state of the knowledge of non-spicular sponges found in carbonates within the fossil record. Also the description of the taphonomic processes are state of the art. Cited literature covers all recently published papers dealing with this topic. Maybe she could give a short notice on Kershaw’s alternative interpretation DOI:10.2110/sedred.2021.1.03.

She should discuss shortly the modern taxonomic framework of the "Keratosa" and notice the two known different chemical compositions of the organic skeletons (papers of Erpenbeck et al. and Ehrlich et al.)

line 10, replace simplest in basic (sponges are not simple...)

line 41 notice also chitin

line 70 what is the proof of "cyanobacteria" there are more types of filamentous bacteria with sheets..

281 Data availability: here a collection number is necessary! And how scientists have access to the material

Fig 2 A may be a better pic should be provided and or an additional one – I see in C and D better portions and in any case she should provide a pic from the margin of the sponge – contact to the cement seen in D

My recommendation is publish this paper soon, it needs only minor revision

Referee #4 (Remarks to the Author):

Comments on Possible poriferan body fossils in Tonian microbial reefs from northern Canada by Elizabeth C. Turner

Does the manuscript have flaws which should prohibit its publication?

No.

If the conclusions are not original, it would be very helpful if you could provide relevant references. The conclusions are original.

Do you feel that the results presented are of immediate interest to many people in your own discipline, or to people from several disciplines?

Yes. These results provide convincing evidence of the earliest animal. This is a key step in understanding the early evolution of life. It will be of wide interest to paleobiologists, geologists, sponge biologists, evolutionary biologists, and everyone interested in the history of life on Earth.

If you recommend publication, please outline, in a paragraph or so, what you consider to be the outstanding features:

The Tonian (~1000-850 Ma) is widely regarded as an interval of significant eukaryote diversification, yet 'Thus far, no convincing animal fossils have been found in the Tonian Period' (Xiao & Tang 2018, *Emerging Topics in Life Sciences* 2 : 61–171). This straightforward, well presented and authoritatively argued article reports Tonian sponges, more than 300 Myr older than any hitherto described. This confirms recent predictions (Lee & Riding 2021, *Geobiology*. 2021, 19:189–198) that keratosan sponges are likely to be found in association with bacterial sediments in Proterozoic rocks. It underscores spongin as a Crown Group character of Demospongiae (Botting & Muir 2018, *Palaeoworld* 27: 1-29).

Additional comments

(i) The Author expertly presents a clear and sufficiently detailed description of the material. She resourcefully marshals cogent arguments to emphasize and discuss the context, significance and interest of this discovery. The writing is clear, straightforward and informative, and to the point. The Author addresses all the questions that I anticipated: geological setting, previous work, calcification, paleoecology, evolutionary implications, environmental context, and broad significance. I unreservedly recommend publication.

(ii) TITLE: Experts will understand Tonian, but the title would be more informative if it included the age of the specimen (890 million years), although I realize this number will not be very precise. For example: Possible poriferan body fossils in 890 million year old microbial reefs from northern Canada

(iii) Lines 46-47: Recent work^{6,7,9,21,22} has proved that vermiform microstructure^{4,5}, an unusual microscopic feature in Paleozoic reefs and stromatolites initially interpreted as protozoan-related^{23,24}, is in fact a keratosan sponge body fossil.

As it stands this statement is correct, but incomplete. Walter (1972, p. 87) compared vermiform structure with algal filaments. Since the Author already refers to Walter (1972) no new reference is required, and this sentence could be rephrased as follows with my suggested insert shown in bold:

Recent work^{6,7,9,21,22} has proved that vermiform microstructure^{4,5}, an unusual microscopic feature in Paleozoic reefs and stromatolites initially interpreted as algal-⁴ or protozoan-related^{23,24}, is in fact a keratosan sponge body fossil.

Author Rebuttals to Initial Comments:

Reviewer 1

Line 34: Bilaterian ‘body’ and trace fossils, as indicated by Chen et al. (2019).

-Added the word ‘body’ as suggested

Line 62-63: Polymud fabric is reported from the same rock that the author studied (Neuweiler et al., 2009). I suggest moving this sentence to the discussion section to support the sponge origin of Little Dal vermiform fabric.

-The material in this paper was controversial at the time and so using it as supporting evidence may be a distraction from the new and different type of evidence presented here. I have elected to leave this comment where it is - as background info rather than supporting info.

Lines 65-75 and onwards: Readers will be interested to know some more details of the reef. I know that the reef was extensively described by the author (Turner et al., 1997, 2000a, b), but it will be good to provide some summary here, as this is a separate manuscript. What are stages 1 and 4, for example? What kind of morphologies can be found there? More illustrations describing the reef growth stage will also be helpful. Figure 3 is a nice model, but it is not supported by the field/slab photo. These can be provided separately in the supplementary material.

-Added and amended text to summarise morphology of framework elements. Also added framework morphology mini-cartoons in Fig 1C. There should be no need to include images of framework types in the paper or appendix because that would be repeating the results of a lavishly illustrated previous study (Turner et al 2000a).

Lines 97-98: It seems to me that there are some remains of calcimicrobes, e.g., Epiphyton or Angusticellularia, in Fig. 2F, although I cannot confidentially identify the structure due to the low resolution of the photomicrograph.

-Unfortunately there are no calcimicrobes that can be assigned to established taxa (or I would have done it long ago!). I never named the filamentous calcimicrobes formally because they just don’t ‘mean’ the same thing as properly formalised taxa – they are variably preserved remnants of cyanobacterial filaments whose exact taxonomic affinity simply cannot be determined – any number of filamentous filamentous colonial cyanobacterial taxa could have produced such a

structure. Although some Soviet-era work (1960s and 70) boldly named Proterozoic calcimicrobes in state-published monographs (although they didn't know what they were, and the idea of 'calcimicrobes' did not yet exist), the taxonomic work was not defensible, and certainly would not be publishable by today's standards.

-I think it is not so much the resolution of the image that is the problem, so much as the preservation of the material that makes the image seem blurry - it's just the way it is, unfortunately. Taphonomic variability of the calcimicrobes was the focus of Turner et al (2000b).

Lines 99-104: It seems that preservation of microfacies 1 is worse than that of microfacies 2 or 3, as vermiform fabric fades away from the center to the margin of Fig. 2C. Also, branching tubules occur along the clast. These characteristics are quite similar to Park et al. (2015, *Sedimentary Geology*, v. 319, p. 124-133) or Lee et al. (2016, *Palaeogeography Palaeoclimatology Palaeoecology*, v. 457, p. 23-30).

-Yes, exactly! I had separate 'results' and 'discussion' sections in this manuscript; the favourable comparison to Lee and Park papers was brought in after the basic features of the Little Dal material had been presented under 'results'. I have not changed this aspect of the paper's structure because it was intended to keep 'results' (bluntly presented with no comparison to other work) and 'discussion' (includes comparison to previous work) separate. Instead, I added a 'background' heading to keep the new data ('results' section) more clearly separate from the background material, and to signal that 'results' are separate from the 'discussion' component of the paper. In other words, I slightly modified the headings to signal that the manuscript's structure resembles that of a standard, long-form scientific paper.

Lines 110-113: Geopetal peloidal accumulations are well figured in Lee et al. (2016).

- This part of the text is in the 'results' section and therefore was deliberately presented without comparison to previous work. Owing to space, a revised figure that explicitly illustrates peloids is now presented as an appendix. The comparison to previous work (including peloids) comes in the 'discussion' section (added Lee et al. 2016 there).

Lines 110-115: Is there any necessity to separate these two paragraphs?

-No. I have now combined them. In fact, the three paragraphs describing the distribution of the vermiform-microstructured masses could be combined into one paragraph (although it might be a bit unreadable).

Line 120: I don't think citing Luo et al. (2016) or Park et al. (2017) are appropriate here. These two references interpret vermiform microstructures as sponges, but except for that, they do not provide any firm evidence that vermiform microstructures have resulted from sponges. Luo et al. (2014) provided 3D reconstructions of the vermiform fabric and compared these with modern keratose sponges, and it is appropriate to be cited here. On the other hand, Lee et al. (2014, *Palaios*, v. 29, p. 27-37, fig. 7a-c) showed that vermiform microstructures can form definitive sponge outlines and is a better reference to be cited here.

-Excellent point that needed to be clarified. I have changed the references cited and added Lee et al 2014.

Line 121: I don't think using apostrophe is necessary when quoting the word keratolite.

-I removed the quotation marks.

Line 137: Poor illumination can be a cause of the absence of cyanobacteria, but I don't think cyanobacteria can be inhibited by high hydrodynamic energy. Phanerozoic calcimicrobes are commonly found in high-energy shoal environments. See Lee and Riding (2021, *Geobiology*) for discussion.

-This comment is maybe generally true for 'normal' dense microbial mats, but (a) the Little Dal calcimicrobes are absent from what is independently known as the highest-energy reef facies (part of stage V reefs), and (b) the calcimicrobial microstructure was extremely porous at a ~10-100 micron scale (there is much more inter-filament space than filaments, actually) and may not have been hydrodynamically robust. I added some text in the background section to explain the framework microbialites and the filamentous calcimicrobial microstructure a little better.

Lines 154-163: One thing to be noted here is that non-spiculate demosponges, including *Keratosa* and *Verongimorpha* (sensu Erpenbeck et al., 2012, *Molecular Phylogenetics and Evolution*, v. 63, p. 809-816), are sister groups to all other spiculate demosponges (Worheide et al., 2012, *Advances in Marine Biology* book, p. 1-78), so maybe primitive sponges could have devoid of spicules. It will be interesting to discuss this idea here, which may be helpful for future studies. Interestingly, *Homoscleromorpha* – the fourth class of sponges – is closer to the *Calcarea* than *Demospongiae*, although *Homoscleromorpha* has siliceous spicules and spongin fibers similar to demosponges which *Calcarea* lack. This may indicate the paraphyletic nature of sponges (*Calcarea*–*Homoscleromorpha* vs. *Demospongiae*–*Hexactinellida*) (Nosenko et al., 2013, *Molecular Phylogenetics and Evolution*, v. 67, p. 223-233), and, possibly that spicules may have appeared more than once during the sponge evolution. If the author has any problem with this comment, please feel free to contact me.

- I agree that Erpenbeck is appropriate here, and added that reference.

Fig. 1C. Please mention that <1005 Ma age is from detrital zircon (Leslie, 2009). If I understood correctly, Milton et al. (2017) analyzed Little Dal Basalt that overlies Little Dal Group (their fig. 2) using U-Pb, not Pb-Pb, and its age is 775 Ma, not 778 Ma. According to Milton et al. (2017, fig. 2), the gabbro dike that the author pointed is connected to the Little Dal Basalt. Please modify this figure accordingly.

-Re age, modified the figure and caption. Given that the dykes and basalt flows are not connected in any known exposure, it would not be accurate to link them in the diagram. The material dated by Milton et al is inferred to be related to the flows, sills, and dykes.

Fig. 2E. I cannot clearly identify geopetal peloidal fabrics here, although I can presume that the fabric does exist. It will be good to provide a better photomicrograph.

-The geopetal peloid accumulations are now depicted in Appendix A, and are labelled with arrows.

Fig. 2F. I can identify vermiform fabrics here, but with provided photomicrograph, it will be hard to recognize vermiform fabrics for non-specialists due to the low resolution of the photomicrograph. I suggest providing a photomicrograph with a higher resolution.

-As with a previous comment, the indistinct nature of the image is not the result of inadequate resolution, but just the nature of the material – not the greatest preservation. I tried to depict the range of preservation quality in Fig. 2 and Extended data Fig. 1 - some of the images are explicitly intended to show what the microstructure looks like when it is poorly preserved, because poor preservation is a reality that people should expect if they go looking for this microstructure in other rocks..

Fig. 3. It will be good if (a), (b), (c) can be marked on the figure. phase5 -> phase 5. I found similar problems a couple of times (e.g., in Fig. 2 caption, line 68, line 119).

- Done. I also revised the text and figs to be more consistent in the way items are numbered: Arabic numerals (1 to 4) for off-reef stratigraphic cycles, upper-case Roman numerals (I to V) for reef stages, and lower-case Roman numerals (i to iii) for microfacies types.

Referee #2 (Remarks to the Author):

The search for a fossil record of metazoans in rocks older than the Ediacaran has had a long, and fraught, history. There have been claims (e.g. Maloof et al. 2010), but none have been upheld to date. Many fossils are now known from the terminal Ediacaran (ca. 560 – 540 million years ago (Ma)) that are almost certainly metazoans, but the precise affinities of nearly all remains problematic. Molecular phylogenies predict that metazoans probably had an origin in the Tonian, and as sponges are sister-group to all other multicellular animals, there has been an expectation that such fossils could be found.

This submission proposes that microscopic vermiform structures, found associated with Tonian microbial reefs ca. 890 Ma, are in fact the remains of keratosan sponges – those with an organic, rather than calcified, skeleton. This is an extremely bold claim. This interpretation is based, in part, on the finding of similar structures intergrown with microbialite (including stromatolitic) reef frameworks that are fairly widespread in the Phanerozoic (e.g. Luo and Reitner, 2016; Lee and Riding, 2021). All Phanerozoic examples are found as dominantly encrusting structures, intergrowing with and attached to, microbialite surfaces. Such vermiform structures can therefore be shown to have grown upwards away from these substrates.

The Tonian structures presented here are proposed to have three expressions: intergrown with microbialite (Fig. 2E,F), within microbialite reef cavities (Fig. 2D), and as reworked clasts (Fig. 2C). But the examples found in reef cavities - which are by far the best preserved - are associated with sediment infill - mainly recrystallized micrite. They are not in any way attached to the reef framework. Most importantly, this sediment infill postdates the precipitation of early marine cements around the reef cavities – see Fig. 2D. So whatever these structures are, they therefore post-dated the active growth of the reef – they are not attached body fossils. Similar claims have been made for vermiform structures within Phanerozoic micrite-filled cavities to be sponges (e.g. Park et al., 2017), but this assertion remain controversial.

-I agree that the relative timing of vermiform mass, geopetal sediment, and marine cement in voids could be considered problematic at first glance, BUT the relative timing of the three items is variable – there are examples in which sediment is first, vermiform second, and cement third AND examples of cement first, vermiform second (with or without detrital carbonate sediment component). This variability in the timing of three void-filling components - cryptic organism, geopetal void-filling detrital carbonate sediment, and marine cement - attests to their penecontemporaneity and to how early all three phenomena were relative to formation of the reef framework. Marine cement is also profuse and abundant in the much smaller pores among the calcimicrobial filaments within the reef framework elements, further attesting to how abundant and early the marine cementation was.

- I am now going into more detail about void-filling materials (I explicitly mention shelter porosity now). I now highlight evidence of both orderings of void-filling items in Fig 2 (i.e., cement first versus cement second).
- The permutations of relative timing of the three void-occluding items (cement, vermiform, and sediment) are already encapsulated in Fig. 3, box ii.
- I have also added some text elaborating on (previously described and published) evidence for the very early timing of marine cement in the reefs.

-It is correct that most of the vermiform masses are not ‘attached to’ (i.e., encrusted on) the reefal microbialites; I did not suggest that they were. The non-encrusting nature of most of the vermiform examples merely indicates that they were not part of the reef frame-building community, but instead were reef-dwellers. In the concluding paragraph, I explicitly referred to the vermiform organism as a ‘reef-dweller’, rather than a ‘frame-builder’.

-Vermiform microstructure in microfacies iii in the Little Dal, however, does ‘encrust’ and ‘grow away from’ microbialite surfaces. I added the word ‘encrust’ to the descriptions of microfacies iiiia and iiib, and improved and expanded the part of Fig. 2 that is dedicated to microfacies iiib, in which ‘encrusting’ is evident.

-The reviewer misunderstood one of the three microfacies – the resedimented clasts contain reef framework that initially grew on the reef, where it locally had a relationship with vermiform-microstructured material. This lithified material was then resedimented as mm- to hm-scale clasts of talus on reef flanks and reef-surface depressions. I do not think that the description of this distribution is unclear in the text. I have revised Fig 3 to include this particular subenvironment explicitly.

Figs. 2E,F and 2C are simply of too low resolution and possibly also poor preservation to really demonstrate that these structures are either intergrown with microbialite, or reworked clasts, respectively. To persuade the reader, far more convincing evidence needs to be presented.

-I cannot do anything about the condition of preservation (good to almost imperceptible); variable taphonomy is always a challenge in deciphering the invertebrate and microbial fossil record, even in Phanerozoic rocks with readily identified body fossils. The Little Dal rocks are almost a billion years old, and they are limestone, which is highly susceptible to all types of diagenetic changes after deposition; it is fortunate that even modest fabric preservation is still present (most Precambrian carbonates are fabric-destructively dolomitised). In geology, the limitations of the rock record are always a factor - we do the best we can with the material that time has left for us.

Wishing for high-quality preservation after a billion years is unrealistic. Even so, the well-preserved material is remarkable.

-I have re-photographed most images in Fig. 2 (providing modest improvement) and provide (a) a new explanatory tracing of a one image that is especially difficult to understand for the reader who is not familiar with carbonate petrology, as well as (b) new higher-magnification images of most of the examples depicted, to highlight the vermiform meshwork and its microscopic distribution.

- The preservation looks bad because it IS bad. The expressly stated intent of Fig 2E (see original caption; images are now in Appendix A) was to depict poorly preserved vermiform microstructure and show that it is spatially and texturally related to geopetal accumulations of peloids – in other words, that poor preservation of Little Dal vermiform microstructure locally yields geopetal peloids. Although this phenomenon is identical to peloid formation in poorly preserved Phanerozoic sponges, I think that for this paper it should suffice just to point it out. The concept exceeds the scope of a brief paper and will be followed up on in later, more detailed publications.

There is much speculation in the Discussion as to the role of oxygen, etc., but this is all rather superfluous. The main claim that these Tonian vermiform structures are indeed sponges needs to be far better illustrated and described, to be believable.

-The discussion is more than mere speculation. The reefs are huge and were constructed by photosynthesisers – it is therefore almost certain that they represented loci of unusually high dissolved O₂ in an otherwise relatively low-oxygen early Neoproterozoic ocean. The concept that early eukaryotes, which require molecular oxygen, may have emerged in the oxygenated vicinity of benthic photosynthetic microbial communities is widespread. Metazoans (animals) have higher oxygen requirements than other organisms. I have combined these widespread pre-existing ideas together with my observations about the distribution of the vermiform masses in the reefs. I think this certainly qualifies as valid and germane interpretation/discussion. It would be remiss not to comment on these obvious implications.

Referee #3 (Remarks to the Author):

Elizabeth Turner presents a very intriguing paper dealing with the very early record (Tonian) of a sponge fossil. Based on molecular reconstructions the origin of animals/sponges should have happened in this time. However, a proof based on fossils was up to now not possible. Also the occurrence of chemical fossils (biomarker), e.g. 24-isopropylcholestane, from this time and a bit later are heavily discussed and problematic. Spicule record of sponges started in the base of the early Cambrian. The basic sponge type was non-spicular and probably related with the “Keratosa” an old taxonomic name of non-spicular sponges with an organic skeleton only. Modern phylogenetic reconstructions have shown that this group is split into the Myxospongia with an organic skeleton formed mainly of chitin and the “Keratosa” with an organic skeleton formed of spongin, a complex proteinaceous material. In any case both types of organic skeletons are very resistant against microbial degradation. In

the end the molds of the skeleton fibers are cemented mainly by carbonate and form “vermiculate” fabrics. These sponges dwell in a wide range of ecological niches and often in dark cryptic areas.

The paper is well written and represents the state of the knowledge of non-spicular sponges found in carbonates within the fossil record. Also the description of the taphonomic processes are state of

the art. Cited literature covers all recently published papers dealing with this topic. Maybe she could give a short notice on Kershaw's alternative interpretation DOI:10.2110/sedred.2021.1.03.

-done

She should discuss shortly the modern taxonomic framework of the "Keratosa"

-I appreciate the intent of this comment, but I do not think that demosponge phylogeny is critical enough to this brief paper to consume any of the manuscript's limited word-count. (The topic is interesting and probably important in the big picture, and will be included in more comprehensive publications to follow.)

and notice the two known different chemical compositions of the organic skeletons (papers of Erpenbeck et al. and Ehrlich et al.)

-done

line 10, replace simplest in basic (sponges are not simple...)

-done (also in abstract)

line 41 notice also chitin

-done

line 70 what is the proof of "cyanobacteria" there are more types of filamentous bacteria with sheets..and cited the specific papers there too

-The cyanobacterial identity of the calcimicrobial filaments was addressed in my previous publications and is not particularly controversial. I do not think that it would be appropriate to rehash the arguments again in this brief paper. Instead, I added some extra explanatory text and referred to the previous papers.

281 Data availability: here a collection number is necessary! And how scientists have access to the material

-The sample numbers of the illustrated material are provided in the figure captions.

-The material is archived in my collection and this is stated in the data availability statement. Because there is no taxonomic component to this paper, there is no type material that requires special archiving.

Fig 2 A may be a better pic should be provided and or an additional one – I see in C and D better portions and in any case she should provide a pic from the margin of the sponge – contact to the cement seen in D

-done

My recommendation is publish this paper soon, it needs only minor revision

Referee #4 (Remarks to the Author):

Comments on Possible poriferan body fossils in Tonian microbial reefs from northern Canada by Elizabeth C. Turner

Does the manuscript have flaws which should prohibit its publication?

No.

If the conclusions are not original, it would be very helpful if you could provide relevant references. The conclusions are original.

Do you feel that the results presented are of immediate interest to many people in your own discipline, or to people from several disciplines?

Yes. These results provide convincing evidence of the earliest animal. This is a key step in understanding the early evolution of life. It will be of wide interest to paleobiologists, geologists, sponge biologists, evolutionary biologists, and everyone interested in the history of life on Earth.

If you recommend publication, please outline, in a paragraph or so, what you consider to be the outstanding features:

The Tonian (~1000-850 Ma) is widely regarded as an interval of significant eukaryote diversification, yet ‘Thus far, no convincing animal fossils have been found in the Tonian Period’ (Xiao & Tang 2018, *Emerging Topics in Life Sciences* 2 : 61–171). This straightforward, well presented and authoritatively argued article reports Tonian sponges, more than 300 Myr older than any hitherto described. This confirms recent predictions (Lee & Riding 2021, *Geobiology*. 2021, 19:189–198) that keratosan sponges are likely to be found in association with bacterial sediments in Proterozoic rocks. It underscores sponging as a Crown Group character of Demospongiae (Botting & Muir 2018, *Palaeoworld* 27: 1-29).

Additional comments

(i) The Author expertly presents a clear and sufficiently detailed description of the material. She resourcefully marshals cogent arguments to emphasize and discuss the context, significance and interest of this discovery. The writing is clear, straightforward and informative, and to the point. The Author addresses all the questions that I anticipated: geological setting, previous work, calcification, paleoecology, evolutionary implications, environmental context, and broad significance. I unreservedly recommend publication.

(ii) TITLE: Experts will understand Tonian, but the title would be more informative if it included the age of the specimen (890 million years), although I realize this number will not be very precise. For example: Possible poriferan body fossils in 890 million year old microbial reefs from northern Canada

-changed ‘Tonian’ to “ca. 890 Ma (Tonian)”. This may be a problem, however, because it contains punctuation (a full stop). Spelling out ‘circa’ instead seems strange. The preferred symbol for ‘approximately’ for geological age is a diacritical (“~890 Ma”), which also is punctuation. I’m a bit stumped.

-removed ‘from northern Canada’ to conform to 75-character limit.

(iii) Lines 46-47: Recent work^{6,7,9,21,22} has proved that vermiform microstructure^{4,5}, an unusual microscopic feature in Paleozoic reefs and stromatolites initially interpreted as protozoan-related^{23,24}, is in fact a keratosan sponge body fossil.

As it stands this statement is correct, but incomplete. Walter (1972, p. 87) compared vermiform structure with algal filaments. Since the Author already refers to Walter (1972) no new reference is required, and this sentence could be rephrased as follows with my suggested insert shown in bold:

Recent work^{6,7,9,21,22} has proved that vermiform microstructure^{4,5}, an unusual microscopic feature in Paleozoic reefs and stromatolites initially interpreted as algal-⁴ or protozoan-related^{23,24}, is in fact a keratosan sponge body fossil.

-added the extra words

Other modifications (not directly linked to reviewers' comments):

- Pruned and reorganised individual words throughout manuscript to diminish word count without changing content.
- Last paragraph (concluding perspective): added the adjectives 'epibenthic and cryptic' for the benefit of the reader.

Acknowledgements

- I inadvertently omitted the main sources of funding in the original version – now corrected.
- *This may also need to be amended in one of the forms I filled in at the time of initial submission (?)*.

Figures:

- The figures are now provided at approximately the anticipated published size.
- Fig 1(C) – Tweak to configuration to convey more accurately the reef – off-reef geometry as established in previous publications.
- Fig 1(C) – Changed reef-stage numbering to Roman numerals to conform to previous publications and preclude confusion with off-reef cycles (Arabic numerals).
- Fig. 2 (incl. caption) – Redone from scratch to improve image quality, help reader understand complex images, and provide more high-magnification images of microstructure.
- Fig. 3 – Added grey line to indicate depth of photic zone (was inadvertently omitted in original version).

Figure captions:

- Fig 3 – Slight modifications for clarity.

Extended data:

- Extended data Fig. 1 - This new appendix consists of an image previously depicted in Fig. 2. The appendix provides higher-magnification insets of two items in microfacies iiib mentioned by reviewers: poorly preserved vermiform microstructure and geopetal peloids.

Reviewer Reports on the First Revision:

Referees' comments:

Referee #1 (Remarks to the Author):

I think the manuscript looks good and it deserves to be published. I only have some minor comments to correct typos.

Line 56. serial-sectioning -> serial-grinding

Fig. 1. Please change -(hyphen) into en-dash (-). U--Pb -> U–Pb, Re-Os -> Re–Os, U-Pb -> U–Pb

Fig. 2h. micromicrobialite -> microbialite. Delete pink line here.

Reference 7 & 31. Lee, J.-Y. -> Lee, J.-H.

Referee #2 (Remarks to the Author):

The revision of this paper presents improved images of vermiform microstructure in Tonian reefs – interpreted here as keratosan sponges.

But I remain unable to support publication of this paper for the following reasons:

1. That vermiform microstructure represents sponges is taken as a given. That it might be something else, e.g. amalgamated layers of compacted peloidal micrite (as proposed by ref. 28), is not fully explored or refuted. Indeed ref. 6 states ‘...[sponge] skeletal fibers and peloidal inter-spaces cannot always be clearly distinguished’, and indeed we are told in this MS text that ‘some vermiform microstructured areas pass laterally to geopetal peloid accumulations in lenticular voids’. The bold claim for the preservation of sponges in the Tonian really must be able to refute alternative explanations for this [in fact rather common] ancient reef microstructure.

2. The best images of vermiform structure are still in reef cavities that postdate early marine cement crust precipitation. This really is critical to the arguments that these structures are sponges. The author states in the response to reviews that, however, the relative timing of the three items is in fact variable and, for example, sediment can be the first phase, vermiform second, and cement third. But I do not see this illustrated in any of the offered figures. In all relevant figures (Figs. 2E,F,J) the micrite (now microspar) sediment cavity-infill postdates the earliest isopachous, cavity-rimming, cement. The caption for Fig. 2F states that ‘Vermiform microstructure occupying lower part of a reef framework void is overlain by pore-occluding marine calcite cement (MC)’ – but this is not true. The thin crust of MC can be seen to predate the micrite infill. The areas labelled MC are in fact mainly later (probably burial) sparry calcite pore-filling cement. The variability of relative timing of the three void-occluding items (cement, vermiform, and sediment) given in Fig. 3, box ii, is simply interpretation, not evidence. The text also states that ‘Void-filling vermiform microstructure of microfacies i and ii commonly either underlies or overlies a thin crust of void-lining fibrous marine cement (Fig. 2e,f)’ and ‘...[vermiform microstructure is] commonly overlain by marine calcite cement that isopachously lines the remaining void space’ – but this is not illustrated anywhere (see above). This statement may be true, but scientists need proof – if the cement is sometimes precipitated first, then unambiguous images to show this must be supplied.

3. Why do these supposed sponges exactly fit the reef crypts, in all their variety of irregular shape and size? This would seem odd – how on earth would they function to pump and filter large volumes of seawater?? Sponges need a lot! Particularly after many cavity pore throats may have been occluded with early marine cement?

I am afraid that I remain unconvinced that these structures are sponges based on the evidence presented.

Referee #3 (Remarks to the Author):

I have no further comments. The author has revised and optimized the early version. References are appropriate and cover all recent significant papers e.g. dealing with fossil "keratose" sponges in carbonates.

The presented paper is a milestone in the understanding of the early fossil record of sponges. It is well written and an important paleontological contribution.

I recommend to accept the paper for publication

Referee #4 (Remarks to the Author):

COMMENTS on revised version of Possible poriferan body fossils in ca. 890 Ma (Tonian) microbial reefs

The revision looks good. I have just a few minor corrections and suggestions.

Lines 140-145: The Author describes interlayering of vermiform microstructure and microbial carbonate:

Lines 140-145: Interlayering of vermiform microstructure with calcimicrobe-free microbialite (microfacies iiii) in the high-energy, well-illuminated reef surfaces of reef stage V supports the interpretation that the vermiform-microstructured organism was not capable of competing with reef-building filamentous cyanobacteria, but instead occupied niches where the filamentous calcimicrobes did not live, owing to (a) poor illumination or (b) high hydrodynamic energy.

Lee & Riding (2021, p. 195 *Geobiology*) drew attention to repetitive interlayering in Late Cambrian Cryptozoön. They suggest that interlayering could reflect repeated cycles of sponge colonization, die-off, microbial growth and sponge recolonization:

Microbial and keratosan layers repeatedly, and relatively thinly, alternate in Cryptozoön (Figure 3). This is unlikely to be fortuitous. In addition

to having similar environmental preferences and behaviours, the microbial mats and keratosans that constructed Cryptozoön could have

been mutualistic in providing substrates, bacteria and organic matter. Keratosan larvae preferentially settle on biofilm, and microbial mats

would therefore offer attractive and stable substrates for sponge colonization (Whalan & Webster, 2014).

In addition, sponges acquire bacterial

symbionts from the surrounding environment (Tout et al., 2017), including bacterial mats (Cleary et al., 2019). On the other hand, dead

sponges would have provided organic-rich substrates for microbial mat colonization. Keratosans are prone to mortality events (Di

Camillo & Cerrano, 2015) due to factors such as bacterial competition (Rützler, 1988), phytoplankton blooms (Stevely et al., 2010), disease

(Easson et al., 2013) and temperature stress (Webster et al., 2008) which is associated with loss of endosymbionts (Cebrian et al., 2011).

Stress due to elevated temperature and salinity may have been common in the shallow water environments occupied by Cryptozoön

during late Cambrian 'greenhouse' conditions (Lee & Riding, 2018). We suggest that frequent alternation of microbial mat and sponge layers

reflects cooperation in Cryptozoön construction as follows:

1. Mats/biofilm surfaces provided sponges with favourable substrates for larval settlement and possibly contributed bacterial symbionts.

2. Keratosans relatively rapidly created extensive enveloping layers but were prone to mortality events that tended to be dome-wide.

3. Microbial mats colonized dead keratosan surfaces, benefited nutritionally

from tissue decay and recreated a substrate suitable for sponge larval settlement.

Repetition of this cycle created Cryptozoön's distinctive alternating and laterally extensive layering.

It would be interesting if the interlayering of vermiform microstructure and microbial carbonate observed by the Author in her specimens similarly indicate that apparently commensal interactions between sponges and microbial mats were already present in the Tonian.

Line 168: Porifera not porifera.

Line 189: Lee, J-H not J-Y.

Author Rebuttals to First Revision:

2021 04 07058A – Author response to second round of review - 2021 06 20

Author responses in blue.

Referee #1 (Remarks to the Author):

I think the manuscript looks good and it deserves to be published. I only have some minor comments to correct typos.

Line 56. serial-sectioning -> serial-grinding
done

Fig. 1. Please change -(hyphen) into en-dash (–). U--Pb -> U–Pb, Re-Os -> Re–Os, U-Pb -> U–Pb
done

Fig. 2h. micmicrobialite -> microbialite. Delete pink line here.
done

Reference 7 & 31. Lee, J.-Y. -> Lee, J.-H.
done

Referee #2 (Remarks to the Author):

The revision of this paper presents improved images of vermiform microstructure in Tonian reefs – interpreted here as keratosan sponges.

But I remain unable to support publication of this paper for the following reasons:

1. That vermiform microstructure represents sponges is taken as a given. That it might be something else, e.g. amalgamated layers of compacted peloidal micrite (as proposed by ref. 28), is not fully explored or refuted. Indeed ref. 6 states ‘...[sponge] skeletal fibers and peloidal interspaces cannot always be clearly distinguished’, and indeed we are told in this MS text that ‘some vermiform microstructured areas pass laterally to geopetal peloid accumulations in lenticular voids’. The bold claim for the preservation of sponges in the Tonian really must be able to refute alternative explanations for this [in fact rather common] ancient reef microstructure.

- Alternative interpretations for vermiform microstructure were extensively explored in previous papers (summarised and cited in lines 60-66), especially Luo & Reitner (2014). The association of vermiform microstructure with confirmed sponge body fossils in the Phanerozoic is now well established (papers cited). Both of these items were outlined in revision 1. The wording in the 'background' section (lines 60-66) has been revised to summarise these previously published comparisons and arguments more explicitly.
- I also now remind the reader of the previous detailed work refuting other possible interpretations in the discussion section (lines 168-170), just to make it clear that that

important work has already been done by others. This is a bit redundant (same material is already presented in background section), but may be worth repeating in the interpretive section.

- Compaction of detrital peloids (ellipsoidal to spherical micrite particles) to produce a micrite groundmass containing complexly anastomosing cylindrical tubules is improbable. This possibility was carefully addressed and dismissed by Luo and Reitner (2014). Conversely, production of peloids taphonomically from decaying sponge tissue has been well documented. The literature sources are summarised in the text.
- The presence of geopetal peloids in Phanerozoic sponge body fossils is well known among fossil reef workers. The means by which sponge tissue is transformed into peloids has been documented (refs already cited). The intimate spatial association of peloids and vermiform microstructure in the LD reefs (where geopetal peloids are otherwise uncommon) is therefore intriguing. The images I selected (now in Extended data Fig 5) highlight the intimate textural and spatial relationship between vermiform microstructure and geopetal peloids at a sub-millimetric scale. The association suggests that geopetal peloid accumulations are a taphonomic product derived through poor preservation of vermiform microstructure in LD reefs, just as they are in Phanerozoic sponges.
- It is incorrect to say that vermiform microstructure is ‘rather common’ – it is common in the Phanerozoic but rare in the Proterozoic. I cited two published examples of demonstrably Proterozoic vermiform microstructure that are accompanied by images that support a comparison to Phanerozoic material. Although Bertrand-Sarfati (1976) mentioned possible latest Neoproterozoic examples, most are not accompanied by references, and the Proterozoic images provided in the one cited reference (Raaben & Zabrodin, 1972) are not comparable to Phanerozoic vermiform microstructure.

2. The best images of vermiform structure are still in reef cavities that postdate early marine cement crust precipitation. This really is critical to the arguments that these structures are sponges. The author states in the response to reviews that, however, the relative timing of the three items is in fact variable and, for example, sediment can be the first phase, vermiform second, and cement third. But I do not see this illustrated in any of the offered figures. In all relevant figures (Figs. 2E,F,J) the micrite (now microspar) sediment cavity-infill postdates the earliest isopachous, cavity-rimming, cement. The caption for Fig. 2F states that ‘Vermiform microstructure occupying lower part of a reef framework void is overlain by pore-occluding marine calcite cement (MC)’ – but this is not true. The thin crust of MC can be seen to predate the micrite infill. The areas labelled MC are in fact mainly later (probably burial) sparry calcite pore-filling cement. The variability of relative timing of the three void-occluding items (cement, vermiform, and sediment) given in Fig. 3, box ii, is simply interpretation, not evidence. The text also states that ‘Void-filling vermiform microstructure of microfacies i and ii commonly either underlies or overlies a thin crust of void-lining fibrous marine cement (Fig. 2e,f)’ and ‘...[vermiform microstructure is] commonly overlain by marine calcite cement that isopachously lines the remaining void space’ – but this is not illustrated anywhere (see above). This statement may be true, but scientists need proof – if the cement is sometimes precipitated first, then unambiguous images to show this must be supplied.

- It is true that many of the best examples of vermiform microstructure post-date cavity-linings of early marine cement, but this is absolutely not the only order of events. The relative timing of the three main void-filling substances (detrital sediment, vermiform mass, and marine cement) truly is variable, which I have now documented in detail in Extended data Fig. 3 (as well as existing parts of in-text Fig 2) in order to satisfy any skeptical reader. In further detail:
 - Reviewer 2 objects (a) to the identification of a pale mass of cement in Fig 2f as ‘marine cement’, and (b) to the same image being used to support the timing of pore-filling vermiform microstructure followed by marine cement. It is difficult to photograph material that has dense, dark components (microbialite; vermiform microstructure) and pale components (cement) in a way that accurately captures the characteristics of both. Illuminating micritic microbial structures adequately in transmitted light (so that they look like more than unintelligible black blobs in photomicrographs) commonly means that cement phases are overexposed. I think that overexposure of cement in order to capture dark microbial masses may explain why reviewer 2 assumed that the pale material labelled ‘MC’ was burial cement (which is typically clear) rather than marine cement (which is always cloudy). To present the material in as convincing a manner as possible, and to make it clear that the material labelled ‘marine cement’ is truly marine cement, I have produced a new Extended Data figure 1. This figure provides enlargements of the isopachous, cloudy marine cement at the pore top and small amounts of blocky, more transparent burial cement that occupies the last small volume of porosity that remained after all of the other phases (sediment, vermiform, and marine cement) had accumulated. The new extended data figure explores all of the void-filling components in magnified insets. This new figure should satisfy even the most skeptical reader that the stated order of pore-filling in this image (sediment followed by vermiform mass followed by marine cement and then very minor burial cement) is accurate.
 - Further comment re reviewer 2 statement that variability in timing of geopetal sediment, marine cement precipitation, and vermiform microstructure have not been demonstrated, and are merely interpretation, not supported by evidence. In the previous revision, I added images (2g-k) showing vermiform microstructure
 - (i) adhering directly to the vertical to overhanging margins of microbialite masses,
 - (ii) interlayered with microbialite, and (iii) at the base of a sediment-filled inter-microbialite space (prior to sediment accumulation). This is evidence for pre-detrital-sediment and syn-microbial-growth timing of some vermiform masses.
 - Collectively, images in Fig 2g-k and Extended data figs depict the following orders of events in the microniches occupied by vermiform masses:
 - in shelter pores and reef-framework voids
 - Marine cement followed by detrital sediment followed by vermiform mass (Fig. 2e; EDFig2)
 - Detrital sediment followed by vermiform mass followed by marine cement (Fig. 2f and EDFig3)
 - in reef-surface depressions
 - Vermiform mass overlain by detrital sediment between microbialite columns (Fig. 2g,h; EDFig4)

- Vermiform mass interfingering with detrital sediment between microbialite columns (Fig. 2g,h,j; EDFig4)
 - in intimate association with micritic microbialite
 - intercalated with microbial masses (Fig. 2g,h,i; EDFig4)
 - encrusting the vertical to overhanging margins of microbialite masses (Fig. 2g,h,k; EDFig 4)
 - I reworded lines 113-114 in the ‘background’ section to be more clear that marine cement is very early, and that its timing relative to geopetal sediment is variable.

3. Why do these supposed sponges exactly fit the reef crypts, in all their variety of irregular shape and size? This would seem odd – how on earth would they function to pump and filter large volumes of seawater?? Sponges need a lot! Particularly after many cavity pore throats may have been occluded with early marine cement?

I am afraid that I remain unconvinced that these structures are sponges based on the evidence presented.

- Sponges filter-feed by creating water currents through canals in their bodies. They are not dependent on ambient water movement - their flagellated collar cells produce vigorous internal currents that deliver suspended organic matter. Phanerozoic cryptic sponges commonly exactly line or completely fill the voids they inhabit –for example, Park et al. (2017) and Luo & Reitner (2016) explicitly depict vermiform masses, interpreted as keratose sponges, that ‘exactly’ fill voids.
- In spite of partly to completely filling voids, cryptic sponges are/were capable of creating fluid-flow through their rock-enclosed bodies, as long as some part of the tissue is exposed to the water column to engage in water exchange. For example, modern endolithic clionid sponges (and their ancient equivalents) fill very complex ‘gallery’ systems (of their own making) in hard substrates such as reefs, shells, and concrete piles and piers (these hard substrates are equivalent to rock). They conduct fluid to the interior of the rock-enclosed gallery system for filter-feeding using very narrow tubes (minute relative to the total volume of the sponge’s rock-enclosed body) that connect the rock-surrounded cavities to the water column (e.g., petrographic images in the widely used 2003 AAPG Memoir 77 carbonate petrography atlas).
- The cryptic habit of some of the vermiform microstructure in the LD reefs does indeed imply that there were cement-free connections between vermiform-filled voids and the water column, out of the plane of the thin section. The size of the connection(s) may have been dwarfed by the total mass of the cryptic sponge, and given the size and complexity of the framework and shelter voids, it is not surprising that such connections may not be evident in every thin section.

Referee #3 (Remarks to the Author):

I have no further comments. The author has revised and optimized the early version. References are appropriate and cover all recent significant papers e.g. dealing with fossil “keratose” sponges in carbonates.

The presented paper is a milestone in the understanding of the early fossil record of sponges. It is well written and an important paleontological contribution.

I recommend to accept the paper for publication

Referee #4 (Remarks to the Author):

COMMENTS on revised version of Possible poriferan body fossils in ca. 890 Ma (Tonian) microbial reefs

The revision looks good. I have just a few minor corrections and suggestions.

Lines 140-145: The Author describes interlayering of vermiform microstructure and microbial carbonate:

Lines 140-145: Interlayering of vermiform microstructure with calcimicrobe-free microbialite (microfacies iiii) in the high-energy, well-illuminated reef surfaces of reef stage V supports the interpretation that the vermiform-microstructured organism was not capable of competing with reef-building filamentous cyanobacteria, but instead occupied niches where the filamentous calcimicrobes did not live, owing to (a) poor illumination or (b) high hydrodynamic energy.

Lee & Riding (2021, p. 195 *Geobiology*) drew attention to repetitive interlayering in Late Cambrian *Cryptozoön*. They suggest that interlayering could reflect repeated cycles of sponge colonization, die-off, microbial growth and sponge recolonization:

Microbial and keratosan layers repeatedly, and relatively thinly, alternate in *Cryptozoön* (Figure 3). This is unlikely to be fortuitous. In addition to having similar environmental preferences and behaviours, the microbial mats and keratosaurs that constructed *Cryptozoön* could have been mutualistic in providing substrates, bacteria and organic matter. Keratosan larvae preferentially settle on biofilm, and microbial mats would therefore offer attractive and stable substrates for sponge colonization (Whalan & Webster, 2014). In addition, sponges acquire bacterial symbionts from the surrounding environment (Tout et al., 2017), including bacterial mats (Cleary et al., 2019). On the other hand, dead sponges would have provided organic-rich substrates for microbial mat colonization. Keratosaurs are prone to mortality events (Di Camillo & Cerrano, 2015) due to factors such as bacterial competition (Rützler, 1988), phytoplankton blooms (Stevely et al., 2010), disease (Easson et al., 2013) and temperature stress (Webster et al., 2008) which is associated with loss of endosymbionts (Cebrian et al., 2011). Stress due to elevated temperature and salinity may have been common in the shallow water environments occupied by *Cryptozoön* during late Cambrian 'greenhouse' conditions (Lee & Riding, 2018). We suggest that frequent alternation of microbial mat and sponge layers reflects cooperation in *Cryptozoön* construction as follows:

1. Mats/biofilm surfaces provided sponges with favourable substrates for larval settlement and possibly contributed bacterial symbionts.
2. Keratosaurs relatively rapidly created extensive enveloping layers but were prone to mortality events that tended to be dome-wide.

3. Microbial mats colonized dead keratosan surfaces, benefited nutritionally from tissue decay and recreated a substrate suitable for sponge larval settlement. Repetition of this cycle created Cryptozoön's distinctive alternating and laterally extensive layering.

It would be interesting if the interlayering of vermiform microstructure and microbial carbonate observed by the Author in her specimens similarly indicate that apparently commensal interactions between sponges and microbial mats were already present in the Tonian.

This line of interpretation is very intriguing, but I am concerned about introducing new discussion items, particularly given the Reviewer #2's concern about some of the interpretations already presented in the paper. I appreciate Reviewer 4's helpful commentary and will take it up in further work to follow.

Line 168: Porifera not
porifera.done

Line 189: Lee, J-H not
J-Y.done